



# Analysis of functional groups in atmospheric aerosols by infrared spectroscopy: sparse methods for statistical selection of relevant absorption bands

Satoshi Takahama[1], Giulia Ruggeri[1], and Ann M. Dillner[2]

[1]ENAC/IIE Swiss Federal Institute of Technology Lausanne (EPFL), Lausanne, Switzerland
[2]University of California – Davis, Davis, California, USA

*Correspondence to:* S. Takahama (satoshi.takahama@epfl.ch)

**Abstract.**

We present an evaluation of four algorithms for achieving sparsity in Fourier Transform Infrared Spectroscopy calibration models. Sparse calibration models exclude unnecessary wavenumbers from infrared spectra during the model building process, permitting identification and evaluation of the most relevant vibrational modes of molecules in complex aerosol mixtures required to make quantitative predictions of various measures of aerosol composition. We study two types of models: one which predicts alcohol COH, carboxylic COH, alkane CH, and carbonyl CO functional group (FG) abundances in ambient samples based on laboratory calibration standards, and another which predicts thermal optical reflectance (TOR) organic carbon (OC) and elemental carbon (EC) mass in new ambient samples by direct calibration of infrared spectra to a set of ambient samples reserved for calibration. We describe the development and selection of each calibration model, and evaluate the effect of sparsity on prediction performance. Finally, we ascribe interpretation to absorption bands used in quantitative prediction of FGs and TOR OC and EC concentrations.

## 1 Introduction

Atmospheric aerosols or particulate matter (PM) can range in size from a few nanometers to tens of micrometers, and exist as complex mixtures of organic compounds, black carbon, sea salt and other inorganic salts, mineral dust, trace elements, and water (Seinfeld and Pandis, 2006). Inhalation exposure of PM can lead to increased morbidity and mortality in susceptible populations; interaction with radiation can lead to visibility reduction and perturbations in the Earth's energy balance; and PM can serve as seeds for cloud droplets and ice particles that lead to additional changes in the climate system (e.g., Dockery et al., 1993; IPCC, 2013).

Fourier Transform Infrared (FT-IR) spectroscopy (Griffiths and Haseth, 2007) is a versatile tool that has been used to detect or measure ammonium, water, ice, mineral dust, organic functional groups (FGs), inorganic ions, and carbonaceous material in laboratory and ambient particles (e.g., Cunningham et al., 1974; Mcclenny et al., 1985; Allen et al., 1994; Cziczo et al., 1997; Hung et al., 2002; Maria et al., 2003; Sax et al., 2005; Hudson et al., 2007, 2008; Liu et al., 2009; Moussa et al., 2009; Day et al., 2010; Hawkins and Russell, 2010; Fu et al., 2013; Takahama et al., 2013; Dillner and Takahama, 2015a). While absorption



bands of isolated molecules culminate in a series of narrow peaks; condensed phase spectra pose challenges for interpretation as these peaks significantly overlap due to heterogeneous broadening of bands from similar bonds vibrating in slightly altered chemical environments (Kelley, 2012). This phenomenon is particularly salient for atmospheric PM, as it comprises a mixture of many different components, with the organic fraction alone consisting of thousands of different types of molecules (e.g.,

Hamilton et al., 2004). Substrate interferences can additionally obfuscate interpretation. For instance, a particular advantage of the FT-IR technique is its capability to directly analyze particles collected on Polytetrafluoroethylene (PTFE, or Teflon) filters, which are routinely used for analysis of gravimetric mass and elemental composition, among other properties. FT-IR can extract a spectrum from an IR beam transmitted through the filter rapidly and non-destructively, without requiring sample pretreatment. In these cases, the PTFE signal can be the dominant component of variation in the spectra (Mcclenny et al.,

1985). In the face of such complexity, statistical approaches are useful in building quantitative models for calibration.

These calibration models can take the form of a multivariate linear equation, in which suitable coefficients are found to combine the effect of absorbances at various wavenumbers of the infrared spectra to reproduce the concentration of a target analyte. Problems of this form are commonly solved by ordinary least squares (OLS) regression, but OLS performs poorly when the system is undetermined (i.e., there are many thousands of wavenumbers and only several hundred samples), and

serial correlation exists among predictor variables (i.e., absorbances among adjacent wavenumbers are not independent of one another) (Weisberg, 2013). Partial Least Squares (PLS) regression is a method that is suitable for obtaining coefficients when such features are present (Wold, 1975; Geladi and Kowalski, 1986; Martens, 1991). The suitability of PLS has been demonstrated in building calibration models for ammonium (Reff et al., 2007), silica (Weakley et al., 2014), organic functional groups (Coury and Dillner, 2008; Takahama et al., 2013; Ruthenburg et al., 2014), and, more recently, organic carbon (OC) and

elemental carbon (EC) reported by thermal optical reflectance (TOR) (Dillner and Takahama, 2015a, b). PLS can be applied to spectra with or without accounting for PTFE contribution a priori. However, we have not yet expounded on how we are able to make accurate predictions from these calibration models, particularly for TOR OC and EC which presumably are composed of a complex combination of molecules.

One approach to facilitate interpretation is variable selection, in which models are reduced to only the relevant wavenumbers

required for prediction (Hastie et al., 2009; Filzmoser et al., 2012; Andries and Martin, 2013). Typically, all wavenumbers are used for prediction in PLS, with irrelevant wavenumbers having small coefficient values. Retaining unnecessary wavenumbers can contribute to overall noise in predictions, degrading model accuracy (Centner et al., 1996; Spiegelman et al., 1998; Höskuldsson, 2001; Reinikainen and Höskuldsson, 2003; Nadler and Coifman, 2005; Cai et al., 2008), and their elimination can additionally make relevant variables more salient for interpretation. Common methods examine various combinations of

variables and their subsets to arrive at the most predictive model (Hastie et al., 2009). However, combinatorial analysis is prohibitive when the dimensionality of the data is large,which is the case for infrared spectra where absorbances for thousands of wavenumbers are available. More efficient methods for variable elimination are accomplished through statistical sampling (Cai et al., 2008; Weakley et al., 2014), but we explore a class of algorithms falling under the domain of sparse methods (Filzmoser et al., 2012; Andries and Martin, 2013). In the context of linear regression, sparsity involves arriving at a set of regression

coefficients in which some or many of the values are exactly zero, allowing the identification of important set of variables





which remain. Sparsity constraints can be imposed in one of several ways in the context of PLS regression, and in this work we explore their merits for analysis of atmospheric PM.

We revisit calibration models for four FGs developed using laboratory standards (Ruthenburg et al., 2014; Takahama and Dillner, 2015), and TOR OC and EC calibration models developed with ambient samples collected in 2011 at seven sites

within the Interagency Monitoring of PROtected Visual Environment (IMPROVE; Malm et al., 1994; Hand et al., 2012) monitoring network (Dillner and Takahama, 2015a, b). We explore four alternative multivariate calibration models that can be built according to different sparsity constraints, and the resulting sensitivity of predictions to sparse formulations. We build two sets of calibration models using two levels of spectral processing (with and without removal of the PTFE interference) for each analyte and algorithm, and further report on the most influential absorption bands in the infrared spectra identified for

prediction of FGs and TOR OC and EC.

## 2   Methods

In this section, we first summarize the experimental protocol detailed by Ruthenburg et al. (2014), in which infrared spectra and reference measurements are acquired (Section 2.1). We then describe the PLS formulation which provides the general framework for solving the calibration problem by projection onto latent variables (LVs), and methods for generating a range

of sparse solutions (Section 2.2). Section 2.3 describes how models are selected and evaluated, and Section 2.4 describes our approach to inferring influential absorption bands from the sparse solutions.

### 2.1   Experimental methods and spectra processing

#### 2.1.1   Laboratory and ambient samples

For this work, we use 794 pairs of ambient samples collected in the IMPROVE monitoring network, 250 laboratory standards,

and 54 blank samples used previously by Ruthenburg et al. (2014), Dillner and Takahama (2015a, b), and Takahama and Dillner (2015) for building FG and TOR OC and EC calibration models with canonical PLS regression. A pair of ambient samples consists of particles collected on 25 mm quartz fiber filters and 25 mm PTFE filters. The quartz filters are analyzed by Total Optical Reflectance (TOR) IMROVE_A protocol for OC and EC mass (Chow et al., 2007), and the PTFE filters are used for acquisition of infrared spectra, among other properties. The 250 laboratory standards consist of 7 compound types

in single, binary, and ternary mixtures, and reference concentrations are obtained by gravimetric analysis of the filters. Four FG calibration models are built for alcohol hydroxyl (aCOH), carboxylic hydroxyl (cCOH), alkane hydrocarbon (aCH), and carbonyl (CO) which comprise these compounds. The blank samples are analytical blanks of PTFE filters analyzed in the laboratory.



### 2.1.2 Infrared spectra

The PTFE filters are scanned (without pretreatment) using a Tensor 27 FT-IR spectrometer (Bruker Optics) with liquid-nitrogen-cooled mercury cadmium telluride detector in transmission mode. Each spectrum is acquired over mid-infrared wavenumbers of 4000 to 420 $\mathrm{cm}^{-1}$, and absorbance spectra are calculated with respect to an empty sample chamber as

reference. The chamber is purged with air free of water vapor and carbon dioxide using a purge-gas generator (Puregas) for all scans.

Two different versions of the spectra described above are used in building calibration models with each described in Section 2.2. Unprocessed ("raw") spectra are unmodified except zero-filled (interpolated) points introduced by the acquisition software are removed such that absorbance values at 2784 wavenumbers at a resolution of 1.3 $\mathrm{cm}^{-1}$ remain. Baseline corrected spectra

are modified according to the procedure described by Takahama et al. (2013). Absorbances below 1500 $\mathrm{cm}^{-1}$ are removed, and the interferences from PTFE are removed by polynomial and linear interpolation between background regions such that the analyte absorption is isolated for analysis. In this process, spectra are interpolated along a wavenumber grid, which is also spaced at a resolution of 1.3 $\mathrm{cm}^{-1}$, such that this spectra type contains 1563 wavenumbers. Both types of spectra have previously shown comparable results for TOR OC and EC prediction with PLS calibration (Dillner and Takahama, 2015a, b).

Example spectra are shown in Supporting Information (SI) Section S1.

### 2.2 Development of calibration models

In multivariate calibration, we seek to solve the linear equation for coefficients $\boldsymbol{b}$ :

$$\boldsymbol{y} = \boldsymbol{X}\boldsymbol{b} + \boldsymbol{e} \, . \tag{1}$$

where $\boldsymbol{X}$ is the spectra matrix (composed by rows of spectra), $\boldsymbol{y}$ is a vector (column matrix) of response values, $\boldsymbol{b}$ are the

regression coefficients (also referred to as the regression vector), and $\boldsymbol{e}$ is a vector of residuals. We continue our discussion under the assumption that $\boldsymbol{y}$ and $\boldsymbol{X}$ are centered by their column means such that an intercept is not included as an additional coefficient. $\boldsymbol{y}$ can alternatively be a multivariate response matrix $\boldsymbol{Y}$, but the univariate case is studied in this work to increase possibility for interpretation (e.g., Haaland and Thomas, 1988; Chong and Jun, 2005). Equation 1 is commonly solved by OLS, where $\boldsymbol{b}$ is found by minimizing the residual sum-of-squares (RSS). However, in spectroscopic applications, the problem

is often complicated by underdeterminacy (many more variables than samples) and collinearity (serial correlation among absorbance values). Therefore, projection onto LVs (e.g., ) or numerical regularization (e.g., Kalivas, 2012) is commonly used to obtain a suitable solution (Hastie et al., 2009). Since four of the five methods used in this work are based on partial least squares or projection onto latent structures (PLS) regression, we first introduce PLS and the underlying structures (loading weights and direction vectors) by which the regression vector is constructed.





### 2.2.1 PLS description

Notation for matrices and vectors are provided in Appendix A. PLS performs a bilinear decomposition and projection of both $\boldsymbol{X}$ and $\boldsymbol{y}$ onto orthogonal bases $\boldsymbol{P}$ and $\boldsymbol{q}$, respectively (Wold et al., 1983, 1984; Geladi and Kowalski, 1986; Mevik and Wehrens, 2007):

$$\boldsymbol{X} = \boldsymbol{T}\boldsymbol{P}^T + \boldsymbol{E}_X$$

$$\boldsymbol{y} = \boldsymbol{T}\boldsymbol{q}^T + \boldsymbol{e}_y$$

The score matrix $\boldsymbol{T}$ relates the two sets of variables and is defined by column matrices of loading weight vectors, $\boldsymbol{W} = [\boldsymbol{w}_1, \boldsymbol{w}_2, \ldots, \boldsymbol{w}_K]$, factor loadings $\boldsymbol{P} = [\boldsymbol{p}_1, \boldsymbol{p}_2, \ldots, \boldsymbol{p}_K]$, and direction vectors $\boldsymbol{R} = [\boldsymbol{r}_1, \boldsymbol{r}_2, \ldots, \boldsymbol{r}_K]$ (ter Braak and de Jong, 1998). The direction vectors can be defined by loading weights and factor loadings:

$$\hat{\boldsymbol{R}} = \hat{\boldsymbol{W}} \left( \hat{\boldsymbol{P}}^T \hat{\boldsymbol{W}} \right)^{-1} .$$

A hat above a symbol denotes the estimator of a variable. From the matrix of direction vectors, we can construct scores and regression coefficients:

$$\hat{\boldsymbol{T}} = \boldsymbol{X}\hat{\boldsymbol{R}}$$

$$\hat{\boldsymbol{b}} = \hat{\boldsymbol{R}}\hat{\boldsymbol{q}}^T .$$

$\boldsymbol{y}$ is therefore estimated as

$$\hat{\boldsymbol{y}} = \boldsymbol{X}\hat{\boldsymbol{R}}\hat{\boldsymbol{q}}^T .$$

The objective function that is satisfied by solutions for $\boldsymbol{W}$ and $\boldsymbol{R}$ are described in Appendix B.

There are two types of variables in our application of PLS regression: the physical variables, or spectral features, corresponding to wavenumbers at which absorbances are measured (columns of $\boldsymbol{X}$), and LVs which represent the underlying components

of the model (columns of $\boldsymbol{T}$). To avoid ambiguity, we will always refer to the latter as LVs. Solutions obtained using PLS with full wavenumber resolution will be referred to as the "full" (wavenumber) solutions.

### 2.2.2 Sparse methods

We consider four methods for obtaining models that require fewer wavenumbers, which are summarized below and described in more detail in Appendix B, and by their respective authors in the cited literature. The underlying principle for wavenumber

selection used in this manuscript is covariance maximization (for PLS regression) or residual minimization (for OLS regression) scaled by an 1-norm penalty ($\| \cdot \|_1$) placed on the regression vector ($\boldsymbol{b}$), weight, or direction vector ($\boldsymbol{w}$ and $\boldsymbol{r}$, respectively). These penalties lead to 1) shrinkage (vector elements tend toward zero), and 2) selection (some elements become exactly zero). One sparse PLS formulation, which we refer to as SPLSa, effectively imposes the 1-norm penalty on the weight vectors $\boldsymbol{w}_k$s of PLS (Lê Cao et al., 2008). The second sparse PLS formulation, which we refer to as SPLSb, imposes the penalty on surrogate





vectors kept in close alignment with the direction vectors $r_k$s (which are closely related to the weight vectors) to target a higher degree of sparsity (Chun and Keles, 2010). Elastic net (EN) regularization is not a variant of PLS but belongs to a separate class of regression algorithms which solve Equation 1. EN generalizes the least absolute shrinkage and selection operator (LASSO; Tibshirani, 1996); the RSS of the model with respect to the response variable is penalized not only by the 1-norm but also the

2-norm applied directly to the regression vector $b$. The primary advantage for including the 2-norm penalty is that this addition imparts a grouping effect, in that variables which co-vary are often selected together rather than one of its members at random. For spectroscopic applications where absorption bands span over several wavenumbers, selection by groups associated with the same absorption band rather than single wavenumbers from each band is desirable for interpretation. The last method of estimation, EN-PLS, is a hyphenated method that combines EN for variable selection and PLS for finding an alternate solution

to Equation 1 using the same subset of wavenumbers as selected by EN. In addition to the number of LVs described above, the sparse methods described above introduce an additional free parameter that controls the magnitude of penalty against lack of sparseness, and different models with varying degrees of sparsity are defined by changing this parameter (Table B1).

## 2.3 Model selection

### 2.3.1 Designation of calibration and test sets

We distinguish between samples used for training and validation of calibration models (the "calibration set") and the test set used for evaluation (the "test set", which has no influence on model development or validation). For FG calibration, 158 laboratory standards are used for the calibration set while 80 similar laboratory samples and all 794 ambient samples are reserved for the test set (Takahama and Dillner, 2015). For TOR OC and EC calibration, the 794 ambient samples are arranged in order of TOR reference concentration and every third sample is selected for the test set and the remaining two-thirds of

samples used for the calibration set (Dillner and Takahama, 2015a, b). Similarly, one-third of blank samples are reserved for the test set while the remaining two-thirds are included in the calibration set. While the division between sets is arbitrary in the TOR OC case, for TOR EC the blanks are first arranged according to their predicted concentrations using a calibration model developed without blank samples (Dillner and Takahama, 2015b) and every third selected for the test set as for ambient samples. Dillner and Takahama (2015b) additionally divided the EC samples into high and low concentration samples to build a hybrid

(piecewise) calibration model for improving predictions for the latter group of samples. For the purpose of this work, we only consider a single calibration model that spans the entire range of concentrations for each algorithm and spectra preparation. No blank samples are included in the FG calibration, though samples with particular FG concentrations of zero (e.g., compounds that only contain other FGs than those for which the calibration model is being built) are included.

### 2.3.2 Metrics for model evaluation

The root mean squared error (RMSE) between the observed ($y$) and estimated values ($\hat{y}$) determined by cross validation (CV) is conventionally used for model selection (Bishop, 2009; Hastie et al., 2009; Arlot and Celisse, 2010). The RMSE given $\theta$ (a



model parameter, or a set of parameters) is defined as

$$RMSE_\theta = \sqrt{\frac{1}{N}\|\boldsymbol{y} - \hat{\boldsymbol{y}_\theta}\|_2^2} \, .$$

This estimate is calculated for $V$ model predictions evaluated against $V$ validation sets to arrive at the RMSE of $V$-fold CV, or RMSECV. To obtain this estimate, the calibration set is divided into $V$ "folds" or subsets; model parameters are trained

on $V-1$ subsets combined together and validated on the remaining subset, with the training and validation sets determined by successive permutation over $V$ repetitions. For this work, samples in the calibration set are arranged in order of increasing concentration for 10-fold Venetian blinds CV in all methods such that the results are deterministic, and the validation sets are likely to be representative of the training sets for each permutation (Dillner and Takahama, 2015a; Takahama and Dillner, 2015). However, as RMSECV generally captures the decreasing model bias and underestimates the growing variance, various

strategies for avoiding overfitting risk has been proposed. One convention is to select a solution within a specified tolerance of the minimum RMSECV — e.g., a fixed value of 10% (Chun and Keles, 2010), or within one standard deviation of the mean as estimated from the CV folds (Hastie et al., 2009). For PLS, Gowen et al. (2011) and Takahama and Dillner (2015) present new metrics which weigh decreasing RMSECV against increasing 2-norm magnitude $\|\boldsymbol{b}\|$ of the regression vector, which is directly correlated with calibration model variance (Faber and Kowalski, 1997). We consider an additional solution defined by

the minimum of a penalized form of the RMSECV (Gowen et al., 2011):

$$pRMSECV_\theta = \frac{RMSECV_\theta - \min_\theta\{RMSECV_\theta\}}{\max_\theta\{RMSECV_\theta\} - \min_\theta\{RMSECV_\theta\}} + \frac{\|\hat{\boldsymbol{b}}_\theta\| - \min_\theta\{\|\hat{\boldsymbol{b}}_\theta\|\}}{\max_\theta\{\|\hat{\boldsymbol{b}}_\theta\|\} - \min_\theta\{\|\hat{\boldsymbol{b}}_\theta\|\}} \, .$$

As $\|\boldsymbol{b}\|$ generally increases with the number of LVs, the model selected with this metric is sensitive to the maximum number of LVs over which the metric is evaluated (Takahama and Dillner, 2015). We evaluate pRMSECVs over the interval of LVs between one and the minimum RMSECV solution in increments of one, leading to selected models generally exceeding 10%

of the minimum RMSECV value and providing a higher degree of parsimony with respect to the number of LVs. Limited justification for this work is provided in Appendix C, and a more dedicated discussion on these metrics is provided by Takahama and Dillner (2015).

Selection of the full wavenumber model is described according to the procedure by Takahama and Dillner (2015), whereby an alternate formulation of the pRMSECV metric is evaluated with an ensemble of penalties and selected by consensus scoring

(Héberger and Kollár-Hunek, 2011). For EN, in addition to the minimum RMSECV solution, we consider a more parsimonious solution within one standard error of the minimum RMSECV. For evaluation and selection of sparse PLS methods, we reduce the complete set of models generated by varying the sparsity parameter (except for EN-PLS, which is fixed by the EN solution) and LVs by considering 1) global minimum RMSECV solutions, 2) solutions within 10% tolerance of the minimum RMSECV model (standard errors of RMSECVs were not readily obtainable), and 3) solutions meeting the minimum pRMSECV criterion.

These solutions are described further in SI Section S2. The final solution for each algorithm and spectra type is selected from among candidate models according to the extent of wavenumber reduction and capability to produce estimates which are evaluated favorably against external reference measurements of TOR OC and EC in the test set (when significant differences in predictions exist). We consider our approach sufficient for this exploratory work, but model selection weighing a metric of





fidelity (e.g., RMSE) against two dimensions of parsimony (number of wavenumbers and LVs) is a potential area of further research.

### 2.3.3 Methods of evaluation

Models are evaluated on sparseness and comparison to reference measurements. Reference measurements include FG abun-
dances in laboratory standards and TOR OC and EC concentrations in ambient samples in the test set. For evaluation of FGs in ambient samples, we sum the carbon mass estimated from FG abundances (which we designate as "FG-OC") and compare against TOR OC. FG-OC is estimated from moles $n$ of FGs as $12.01\,\mu\text{g/mole} \times (0.5\,n_{\text{aCH}} + n_{\text{cCOH}})$ (Ruthenburg et al., 2014). We characterize sparseness by the number of selected wavenumbers, or non-zero variables (NZVs). We use the Pearson's cor-
relation coefficient ($r$) to determine how closely related the predictions are (as characterized by linearity), and the regression
slope determined by orthogonal regression (also referred to as major axis regression) to characterize overall bias between two sets of values (Ripley and Thompson, 1987; Wittig et al., 2004). The intercept from the regression is not reported to simplify the discussion (it is generally near zero); for detailed evaluation that also considers detection limits and values close to zero, a further study and exposition is recommended. Orthogonal regression is appropriate for our comparisons as it considers that both concentrations being compared are subject to error. However, we do not weigh each observation by the magnitude of
their errors, as measurement errors are currently not well characterized for each sample and erroneous weighting can lead to mischaracterization of bias.

### 2.4 Interpretation of influential absorption bands

While examining a sparse set of regression coefficients are informative for identifying important absorption bands, interpreta-
tion is still complicated by the compensation of interfering bands (Haaland and Thomas, 1988; Kvalheim et al., 2014). Rather
than viewing these coefficients directly, we examine the loading weights of PLS to aid in this interpretation. The first weight component $\boldsymbol{w}_1$ of PLS can indicate a first approximation to the "pure component" representation of the the response variable, but may be obfuscated when large contributions to the signal come from components that are not the target analyte (Haaland and Thomas, 1988). This can be especially true for our models where the spectra are not baseline corrected to remove the PTFE signal prior to calibration. Therefore, we also examine the Variable Importance in Projection (VIP) metric (e.g., Wold, 1993;
Chong and Jun, 2005), which considers normalized loading weights with the fraction of captured response, and summarizes the importance of each wavenumber $j$ for $k$ LVs:

$$VIP_{jk} = \sqrt{M \sum_{h=1}^{k} \left( \frac{SS_h}{\sum_{h=1}^{k} SS_h} \right) \left( \frac{w_{jh}}{\|\boldsymbol{w}_h\|_2} \right)^2} \quad \text{where} \quad SS_h = q_h \boldsymbol{t}_h \boldsymbol{t}_h^T \ .$$

VIP is an expression of the normalized loading weight $\boldsymbol{w}$ of the $k$th LV weighted by the corresponding fraction of captured response (sum-of-squares, or $SS$). The average of squared VIP scores across wavenumbers equals one ($1/M \sum_{j=1}^{M} VIP_{jk}^2 = 1$),
so a value of VIP greater than unity is often taken to be an indicator of an important variable. However, this criterion is not a strict one, and determination of useful thresholds is dependent on the proportion of unimportant variables, correlation





among important variables, and variation in coefficient strengths present in the data set (Chong and Jun, 2005). We discuss our interpretation and selection of threshold in Section 3.3.

For TOR OC and EC, we further provide an additional level of qualitative interpretation by associating absorption bands of vibrational modes to FGs that contribute to our capability for predicting TOR OC and EC. For this purpose, we examine regression coefficients alongside VIP scores and consider that negative coefficients a) compensate for positive artifacts in other absorption regions, or b) are themselves artifacts of oscillations that occur in regression coefficients when the number of LVs in the model is large (Gowen et al., 2011). In the latter case, we consider its influence alongside the positive contributions of adjacent wavenumbers. Wavenumbers and vibrational modes of organic bonds tabulated by Shurvell (2006) and Pavia et al. (2008) have been used in this analysis. We note that carboxylates and aminium FGs have different vibrational frequencies compared to their neutral forms. As our PM samples are analyzed under dry conditions (Ruthenburg et al., 2014), ionic forms of carboxyl and amine groups are not expected in significant quantities. These FGs can have similar vibrational frequencies in crystalline structures at ambient conditions (e.g. L-alanine, from Caroline et al., 2009), but have not been considered in the interpretation of the different solutions discussed in this work. As organic PM mass estimated by the FT-IR technique without consideration for these structures often agrees with OM reported by other analytical techniques (e.g., Russell et al., 2009; Gilardoni et al., 2009; Corrigan et al., 2013), we expect that their contribution to OC quantification may also be small.

## 3    Results and Discussion

In this section, we first describe the range of sparse models that are generated by different algorithms and tuning parameters, and present the models selected based on validation within the calibration set (Section 3.1). We then discuss our evaluation of these models on the test set samples, which are samples excluded from the model building and selection stage (Section 3.2). We conclude with a discussion of our interpretation of absorption bands used by these calibration models (Section 3.3).

### 3.1    Sensitivity of wavenumber reduction to sparse formulation

The sensitivity of RMSECV to models formulated with different NZVs are shown in Figures 1 and 2 for raw and baseline corrected spectra, respectively (FGs are shown in fixed order from highest to lowest wavenumber of absorption bands — aCOH, cCOH, aCH, and CO — in all figures). In general, we note that decreasing the number of NZVs does not necessarily reduce prediction quality as assessed through the RMSECV. The solution selected is also indicated in each panel and summarized in Table 1. For TOR OC and EC, the most parsimonious solution with respect to NZVs and LVs within 10% of the minimum RMSECVs are selected for the SPLSa and SPLSb, the solution within one standard error above the minimum RMSECV for EN, and the minimum pRMSECV for EN-PLS are chosen. The selection criterion for FGs varies by method and spectra type, and a more detailed evaluation is described in SI Section S2. Additional consideration is required as RMSECV indicated for laboratory standard spectra may not necessarily reflect the prediction error when extrapolated to ambient sample spectra (Takahama and Dillner, 2015), and predictions of FGs cannot be evaluated individually as no reference measurements exist for these samples. Therefore, while candidate solutions are formulated with respect to the minimum RMSECV and pRMSECV,





we select one from among them after considering agreement (correlation and regression slope) of the combined FG-OC with TOR OC, and overall reduction in NZVs.

We observe from a comparison of methods that the range of RMSECVs estimated by each model algorithm depends on the response variable (FGs or TOR OC and EC) and spectra type, and none consistently outperforms the rest. EN-PLS is

able to achieve lower apparent RMSECVs than EN in many cases even while using the same wavenumbers, though this difference may partially be due to underestimation of the final RMSECV resulting from awareness of validation samples in the PLS stage of model evaluation (Appendix C). However, Lee et al. (2011) also suggests that the ability of EN-PLS to access lower dimensional spaces leads to more accurate calibration models than EN (i.e., further transformations are applied, and unnecessary information is discarded by appropriate LV selection). Imposing sparsity on individual PLS weight or direction

vectors as targeted by SPLSa and SPLSb eliminates different wavenumbers for each LV; when combined over large number of LVs that we have in our models (Table 1), this approach does not guarantee overall sparsity in the final regression coefficients. The tuning parameter for EN controls the sparsity of the NZVs directly through the regression vector rather than the PLS direction vectors, thereby permitting more control over the range of NZVs than SPLSa or SPLSb. This control can allow construction of more sparse solutions, though for extreme reductions in NZVs we observe consistently high RMSECVs. SPLSb

was formulated to achieve higher degrees of sparsity (fewer NZVs) by penalizing the surrogate of the direction vector, rather than the direction vector directly. In the case for TOR OC and EC where the identical model selection criterion is used, we find that our selected SPLSb solutions are more sparse than SPLSa in 3 out of 4 scenarios, with an exception noted for the TOR EC calibration model using baseline corrected spectra.

The reduced wavenumber models using raw spectra resulted in fewer NZVs than using baseline corrected spectra for 14 out

of the 24 of the cases examined, indicating that it is possible for sparse methods to effectively remove the PTFE interference and achieve suitable performance. One potential explanation may be that isolated regions of PTFE interference can be used efficiently to correct for the remaining interferences from the analyte regions. On average, NZVs for both spectra types are reduced by approximately 20% for the solutions chosen. However, reductions can be as low as 1–9% for any substance, mostly achieved by EN (with the exception of CO for the baseline corrected case, where the fewest NZVs is achieved by the SPLSa

algorithm). The highest percent reduction is achieved for aCH (99%; corresponding to 40 NZVs) using raw spectra with EN, which is a surprising result given the richness of features in the aCH region (Guzman-Morales et al., 2014). Forty NZVs corresponds to the fewest in the set evaluated; tied with CO of baseline corrected spectra. Takahama et al. (2013) previously found that molar absorption coefficients for carboxylic and ketonic CO were approximately similar for the compounds used in their study. If a similar conclusion holds for carboxylic, ketonic, and ester CO absorption in the compounds used in this study,

it is plausible for such a reduced PLS model to result if a subset of wavenumbers common to these three types of CO bonds is selected. Comparing the minimum NZVs obtained in this study, the TOR OC and EC retain more than 110 each, where NZVs for individual FGs are all below this value. This is consistent with our understanding that OC and EC comprise complex mixtures beyond a single FG. However, the NZVs for TOR OC or EC are less than the sum of individual functional groups; not all of the NZVs from these individual FGs are necessary for prediction of OC and EC.





## 3.2 Sensitivity of predictions to model sparsity and evaluation against reference measurements

Prediction of FG concentrations in laboratory samples shows good agreement with laboratory samples with $r > 0.9$ and slopes within 5% of unity (SI Section S2.2.2). The FT-IR FG-OC estimated with the full models show high correlation with TOR OC ($r > 0.9$) (Figure 3). Estimates of FG-OC from the raw spectra calibration model exhibit lower bias on average with a

regression slope of 0.97, while the regression slope is 0.75 with the baseline corrected model. For estimates from SPLSa and SPLSb models, correlation with TOR OC is mild to strong ($r > 0.7$), with regression slope varying between 0.82 and 1.69. However, these two metrics do not fully capture the bifurcating relationships between predicted and reference concentrations as shown by the scatter plots (e.g., for SPLSa, raw spectra model and SPLSb, baseline corrected model) which speak to the way in which predictions from the reduced models deviate from those of the full models for different types of samples. For

instance, the moles of aCH predicted by SPLSb for rural samples are more than a factor two higher than that predicted by the full model, while the agreement is within 10% for urban samples (Figure 4). Using the same wavenumbers as EN, EN-PLS predictions are more consistent with those of the full model and exhibit generally higher correlation with the reference TOR OC concentrations ($r = 0.92$ and 0.97 compared with $r=0.87$ and 0.9 for the raw and baseline corrected models, respectively). Examining the correlation for each FG reveals that the major difference is in the estimated aCH by EN and EN-PLS; the aCH

predictions especially for EN in urban areas are 1.5 times higher than the full solution, largely contributing to an increase in the slope from 0.97 to 1.24. In contrast, the EN-PLS solutions predict aCH within 10% of the full model.

It is worth noting that all sparse models predict higher concentrations of aCOH in both urban and rural samples than full models on average by a factor of 3–8, when raw spectra are used for calibration. For this spectra type, urban cCOH samples are also on average over-predicted by a factor of 4 (except by SPLSb). Predictions for CO generally exhibit less variation. However,

as the aCOH and cCOH are not as large contributors to the OM as aCH (60–70% of OM mass according to the full model; Ruthenburg et al., 2014), this is not prominently reflected in the comparison against TOR OC. There is less variability in aCOH and cCOH according to sparse formulations using baseline corrected spectra, suggesting that removal of PTFE interferences in this region may be relevant. We provide a limited illustration of how predictions vary in the baseline corrected models according to selected wavenumbers in Section 3.3.1.

In comparison to the FG-OC predictions which are extrapolated from laboratory standards to the composition domain of atmospheric OM, predictions for TOR OC and EC made by direct calibration to ambient samples show remarkable consistency with the full model solution (Figure 4), and capability for accurate prediction with respect to evaluation TOR OC and EC measurements (Figure 5). The difference with respect to the full model solution is generally within 30% with highest differences observed in rural samples, but this is likely due to the lower concentrations in these areas. Comparing against TOR OC test set

samples, $r \geq 0.98$ and slope within 8% of unity except for EN; for TOR EC, $r \geq 0.93$ and slope less than 5% of unity, again except for EN. EN predictions retain high correlations but slopes with respect to reference are between 0.89 and 0.95 for both TOR OC and EC.

We have only compared one possible solution from each method, but other solutions can be generated for a given algorithm by changing the model parameters; there may be possible solutions which are better suited. However, as concluded previously



(somewhat obviously) in PLS applications to aerosol FT-IR spectra, predictions are most robust when samples in the evaluation set are similar to those in the calibration set (Dillner and Takahama, 2015a, b; Takahama and Dillner, 2015), and this also applies to sparse calibration models. Calibration models developed with laboratory standards and ambient samples predict concentrations in laboratory standards and ambient samples, respectively, with only mild sensitivity to model formulation.

Largest variations in predictions occur when extrapolating from laboratory standards to ambient samples.

## 3.3 Influential absorption bands

VIP scores and the sign of regression coefficients at each wavenumber are shown in Figures 6 and 7 for raw and baseline corrected PLS model solutions, respectively. As EN-PLS uses the same wavenumbers as EN with the same or better performance metrics (Sections 3.2), we omit discussion of EN in this section. We first confirm that FG calibration models use wavenumbers

which are consistent with our physical understanding of the vibrational modes belonging to FG groups (Section 3.3.1), but also include PTFE interferences and spurious correlations with other absorption bands. For describing our interpretation of bonds and FGs that give rise to our capability to predict TOR OC and EC, we begin with the EN-PLS solution (Section 3.3.2) as it is the most parsimonious subset of each of the full and other sparse solutions, and then extend our interpretation to the remaining solutions (Section 3.3.3).

### 15 3.3.1 Interpretation of FG solutions

We first describe our interpretation of the most parsimonious EN-PLS solution for each FG, and extend our interpretation to the SPLSa, SPLSb and full spectra solutions. For aCH, CO, and cCOH FGs, the same vibrational modes are used for both baseline corrected and raw spectra solutions. In the case of aCH, the C-H stretching mode (near 2900 $cm^{-1}$) and what appears to be spurious correlation with C=O stretching (near 1700 $cm^{-1}$) in carbonyl compounds are found. In the calibration experiments,

carbonyl compounds were included in the set of compounds used as laboratory standards for the calibration of the aCH FG (i.e. malonic acid, adipic acid, suberic acid, arachidyl dodecanoate and 12-tricosanone; Ruthenburg et al., 2014). For CO, the C=O stretching is used. The C=O stretching vibration is used also in the cCOH calibration, in both the baseline corrected and raw spectra solutions. Different sections of the spectra in the region of C=O stretching absorption are used for CO and cCOH solutions, though they do not necessarily coincide with wavenumbers expected for their specific chemical environments.

Carbonyl in carboxylic acids are nominally centered around a lower vibrational frequency ($\sim$1710 $cm^{-1}$) than in esters ($\sim$1735 $cm^{-1}$), but in our solution the wavenumbers used by the cCOH model in the carbonyl region is higher than that for the CO model which also included esters in the calibration set. While carbonyls have a relatively narrow absorption band compared to O-H stretching used by alcohol (3500–3200 $cm^{-1}$) or carboxylic hydroxyl groups (3400–2400 $cm^{-1}$), it is broad enough such that EN-PLS selects different regions of the band that may not correspond to the location of peak absorbance. The high VIP

scores and negative coefficients found for the wavenumbers around 2360 $cm^{-1}$ in both CO and cCOH raw spectra solutions can be associated with the background (PTFE) correction which can interfere with carbonyl quantification, which lies at the shoulder of the C-F stretching mode of PTFE. Additional wavenumbers associated with high VIP scores found in the cCOH in the raw solution is also attributed to background correction. In the case of aCOH, different vibration modes are used by





the baseline corrected and the raw spectra solutions. While the baseline corrected solution uses the alcohol O-H stretching mode (near 3300 $cm^{-1}$), the raw spectra solution uses C-O-H bending (680–620 $cm^{-1}$) and C-O stretching vibrational modes (1200–1015 $cm^{-1}$). In the baseline corrected solution for aCOH, the high VIP score at 1707 $cm^{-1}$ is interpreted as a spurious correlation with CO, present in compounds in the calibration set. The high VIP scores in the aCOH raw spectra solution are

attributed to background correction.

SPLSa, SPLSb and full spectra solutions use the same vibrational modes of the EN-PLS solution for the quantification of cCOH, aCH and CO. The additional peaks in the raw spectra solutions are interpreted as being associated with background correction. For aCOH, the less parsimonious methods use the O-H stretching in both the baseline corrected and the raw spectra solutions in contrast to the EN-PLS solution, which uses two different vibrational modes for different spectra types.

Similarity in predicted abundances is not necessarily anticipated by the number of wavenumbers used. An illustration is provided in Figure 8, where a group of similar baseline corrected ambient spectra is overlaid on VIP scores for the full model, SPLSa, and EN-PLS. While the EN-PLS solutions have a higher degree of sparsity than the SPLSa (2–6% of original wavenumbers EN-PLS compared to 6–93% for SPLSa) for every FG except baseline corrected CO (Table 1), EN-PLS estimates are more closely aligned with the full wavenumber predictions. The aCOH abundance of sparse solutions are an anomaly in that

they are correlated with each other and overpredict the full wavenumber estimates by a significant amount (which is true for all sparse solutions; Section 3.2). This pattern may be the result of selecting narrow bands of wavenumbers for the estimation of this FG, when O-H stretching from hydroxyl groups in alcohol compounds exhibit a broad absorption band (Pavia et al., 2008). Even for this similar group of spectra which presumably share similar chemical composition, the varied patterns in predicted concentrations across sparse models indicate that the features selected for quantification by laboratory standard calibrations

may not be consistent with additional features present in ambient samples. We anticipate that this analysis of such sensitivity can further guide the selection of laboratory standard mixtures for calibration, in addition to measurement intercomparisons.

### 3.3.2 Interpretation of TOR OC and EC solutions from EN-PLS

For both the prediction of TOR OC and EC, different sets of wavenumbers (i.e., absorption bands) are used between the raw (Figure 6) and baseline corrected (Figure 7) spectra solutions. In the raw solution for TOR OC and TOR EC prediction, large

VIP scores with negative coefficients near 2000 $cm^{-1}$ and 1200 $cm^{-1}$ constitute a means for the correction of the PTFE absorption and scattering. For instance, this correction compensates for the positive artifact near 4000 $cm^{-1}$ where the PTFE scattering is the only contributor to the infrared signal, but PTFE contributions are also present in regions of analyte absorption. As the largest VIP scores are associated with PTFE correction and smaller VIP scores associated with wavenumbers in absorption bands of target analytes, we primarily consider absorption bands above a low VIP threshold of approximately 0.5

in this work.

Even by excluding wavenumbers with extremely small VIP scores and PTFE contributions, many interpretations for contributing FGs still exist on account of the large number of overlapping absorption bands at each wavenumber used by the two spectra types. In this first analysis of relevant wavenumbers for TOR OC and EC calibration, we present our interpretation through a "common FG hypothesis" in which we assume it most likely that predictions by the raw and baseline corrected





spectra models are primarily enabled by a common set of FGs. This framework leads to the possibility that the same FG may be used by the two solutions by means of different vibrational modes (at different wavenumbers), and the inference that these comprise essential FGs necessary for prediction. We cannot exclude the possibility that there exists a suite of FGs with approximately similar capability to provide, in some combination, quantitative prediction of TOR OC and EC, leading to two

models that require less than maximal overlap in FGs. If we consider an extreme case which we denote as the "divergent FG hypothesis," a pair of models may use a minimally redundant set. This approach to band assignment can lead to an intractable number of possibilities, and is also considered less plausible given the similar level of accuracy and robustness attained by the two models. Table 2 presents our findings of bonds found in each model; additional possibilities for each wavenumber are documented in Section S3. We limit our discussion below to main FGs found by both models through the perspective of the

common FG hypothesis.

    We preface our interpretations that at this time, we make no claim regarding the relative contributions of each FG to TOR OC or EC mass, as our VIP analysis considers the importance of spectra absorbances (and not FG abundance) to the mass concentrations. Knowledge regarding molar absorption coefficients and the relationship of FG to carbon abundance (Takahama et al., 2013) are additionally necessary to relate absorbance with the mass concentrations to which the models are calibrated;

this information is unavailable and not possible to estimate unambiguously from the set of regression coefficients for these complex mixtures.

    The wavenumbers used by the TOR OC calibration models correspond to vibrational modes associated with major FGs of organic PM. Carbonyls associated with carboxylic acids, ketones, aldehydes, and esters are used by both models through the C=O stretch (1700–1750 $\text{cm}^{-1}$). Carboxylic acid groups are inferred through O-H and C=O stretch by the baseline corrected

model and O-C=O bending in the raw spectra model. The alcohol FG is used by both baseline corrected and raw solutions but by means of different vibrational modes (i.e. O-H stretch in baseline corrected spectra solution and C-O-H bending in the raw spectra solution) as in the case of the EN-PLS solution for aCOH FG. N-H bending associated with amide and amines (1640–1550 $\text{cm}^{-1}$) is used by both models, with a strong N-C=O (630--570 $\text{cm}^{-1}$) and C=O out-of-plane bending absorption in amides (615--535 $\text{cm}^{-1}$) additionally used by the raw spectra solution. Given the additional support for assignment of

the N-H bending mode to amide in the raw spectra solution our common FG hypothesis favors the interpretation that this mode in the baseline corrected spectra mode may also be more strongly linked with amides. While not currently reported in measurements of organic aerosol by FT-IR (e.g., Russell et al., 2011; Ruthenburg et al., 2014), amide-containing compounds have been suggested to partition to the aerosol phase as well as formed through condensed phase reactions (Pitts, Jr. et al., 1978; Barsanti and Pankow, 2006; Murphy et al., 2007). Various vibrational modes associated with aromatic and alkene species have

medium or weak absorption in regions used by both raw and baseline corrected spectra solutions. The modes associated with conjugation of C=O with phenyl and alkenes absorb in regions used by both baseline corrected and raw solutions. Alkane chains are used by the baseline corrected solution by means of the C-H stretching mode in the wavenumber range 2913–2921 $\text{cm}^{-1}$. In the raw solution, the assignment is less clear, but region of 1505–1517 $\text{cm}^{-1}$ used by the model (with high VIP scores and positive coefficients) overlaps with the shoulder of $CH_2$ bending vibrations of alkane chains near 1475 $\text{cm}^{-1}$. Given

the large contribution of alkane C-H groups to the overall organic aerosol mass estimated as estimated by FG calibrations





(60–70%; Ruthenburg et al., 2014), the lack of a more obvious correspondence between regression coefficients and vibrational models of saturated CH groups is unexpected.

The baseline corrected TOR EC calibration model appears to rely on similar absorption bands and FGs as TOR OC; this can be partly explained by the restricted range of wavenumbers used. However, as the regression coefficients are different we

note that the bands are weighted differently in arriving at their respective predictions, possibly indicating their use for OC artifacts in quantification of TOR EC. Many similarities in the structure of VIP scores with the raw solutions of TOR OC can be accounted to the PTFE corrections previously described, though the main spectral features used by the raw spectra TOR EC model appears to be vibrational modes in the molecular fingerprint region that overlaps with the absorbance from the C-F stretching of PTFE ($\sim$1200 $\mathrm{cm}^{-1}$) and the adjacent region between 1497–1531 $\mathrm{cm}^{-1}$, which is at the lower boundary of the

baseline corrected solution.

Both TOR EC models may use four FGs used by the TOR OC solutions: aromatic and ring structures, amines and amides, and esters. The C-C ring stretch is present in the baseline corrected solution at $\sim$1600 $\mathrm{cm}^{-1}$ and $\sim$1500 $\mathrm{cm}^{-1}$ for the raw spectra solution. In the former case, there may also be indication for the conjugation of phenyl rings with carbonyl compounds in ketones, aldehydes, and esters and weak absorption due to overtones in substituted benzene rings. Amines and amides

may be incorporated through the N-H bending absorption by the baseline corrected solution (1587–1601 $\mathrm{cm}^{-1}$) and the C-N stretching in aromatic amines in the raw solution. Additionally, given the positive regression coefficients, it is possible that the TOR EC raw spectra model uses the fingerprint region (1100–1300 $\mathrm{cm}^{-1}$) associated with vibrational modes of amines and ethers; not only using this region compensating for the possible artifacts due to PTFE absorption as for TOR OC. The C=O stretching vibration (1722–1739 $\mathrm{cm}^{-1}$) in the baseline corrected solution can be attributed to ketones, aldehydes, or esters, but

when harmonizing with the raw spectra solution according to the common FG hypothesis, the ester assignment through the C-O-C antisymmetric stretch (1275–1279 $\mathrm{cm}^{-1}$) is considered possible. Ester formation has been associated with aqueous-phase processing of biogenic organic compounds in atmospheric PM (Surratt et al., 2007; Kroll and Seinfeld, 2008), so its association with EC is unexpected and tentative.

The band assignments for TOR OC are perhaps not surprising given that many of the FGs have been used previously for

quantification of organic PM, but it is worth considering our interpretation for TOR EC in context as FT-IR is not commonly employed for the study of elemental carbon or similar substances. "Elemental carbon" strictly refers to $\mathrm{sp}^2$ carbon not bound to other elements and has the property of thermal stability with respect to vaporization up to 4000 K in an inert atmosphere, or 340 °C in an oxidizing environment (Petzold et al., 2013; Lack et al., 2014). EC by TOR is therefore quantified by heating PM samples at these high temperatures in the presence of oxygen, and its quantity separated from the preceding OC vaporized

anoxically by an operationally defined protocol based on evolving filter optical properties (Chow et al., 2007). Atmospheric elemental carbon as reported by this method is likely to represent a set of strongly light-absorbing, low-volatility compounds that characterize carbonaceous material formed or emitted from combustion processes, rather than in one of the chemically pure allotropic forms (e.g. graphite and diamond) (Chow et al., 2004; Petzold et al., 2013).

Direct measurements of elemental carbon and associated substances by infrared spectroscopy are not numerous, but Friedel

and Carlson (1971, 1972) recorded infrared spectra of ground graphite and coal; reporting strong, broad absorption bands





centered near 1600 $cm^{-1}$ superposed on a wider band between 1800–900 $cm^{-1}$ in both substances. While 1600 $cm^{-1}$ is near reported lattice frequencies for crystalline graphite (Tuinstra and Koenig, 1970), the band becomes visible only with extensive grounding of this material in which crystalline structure is lost (Friedel and Carlson, 1972). Therefore, Friedel and Carlson (1972) attribute these bands to non-crystalline graphite structure, presumably carbon-carbon bonds which occur in exposed

edges of the ground graphite (Szabó et al., 2006). The 1600 $cm^{-1}$ band has in the past been attributed to aromatic or carbonyl structures as their resonances overlap in this region, but the absorption lineshapes of these two structures are not accompanied by the broader band spanning a range of 900 $cm^{-1}$ (Friedel and Carlson, 1972; Szabó et al., 2006; Stankovich et al., 2006; Si and Samulski, 2008). The absorption band around 1600 $cm^{-1}$ can also be found in spectra of polycyclic aromatic hydrocarbons such as anthanthrene and benzo[ghi]perylene and has been attributed to the stretching of the aromatic C=C bonds (Karcher

et al., 1985).

    The absorption band around 1600 $cm^{-1}$ is required by our baseline corrected solution and the raw solution appears to use a shoulder of the same band at lower wavenumbers (range 1497–1531 $cm^{-1}$). The relevance of this band to both solutions for the prediction of TOR EC is consistent with the presence of $sp^2$ bonds in ring-structured substances (as discussed above) known to be emitted from combustion sources (Flagan and Seinfeld, 1988; Bond et al., 2004). We attribute the absorption around 1700

$cm^{-1}$ in the baseline corrected solution to (the shoulder of the) ester C=O stretch for harmony in interpretation with the ester C-O-C antisymmetric stretch at 1275 $cm^{-1}$ in the raw solution, as described previously. Because of the complexity of the PM mixture captured by the infrared spectra, we are unable to identify the broader feature of 1800–900 $cm^{-1}$ associated graphitic structure reported by Friedel and Carlson (1972). However, as the baseline corrected spectra solution does not use information below 1500 $cm^{-1}$, it appears that accurate calibration models of TOR EC can be constructed even by omission of a large

portion of this broad band. Amines have been associated with anthropogenic emissions and strong partitioning behavior to the aerosol phase (e.g., You et al., 2014), so identification of this FG is plausible given our understanding of EC sources.

    We explicitly remark that while the absorption bands discussed in relation to VIP scores are all observed in the overall infrared spectra used for building calibration models, they are associated with the PM mixture and do not correspond to direct observations of bands in physically or chemically isolated specimens of TOR EC. The bands are selected mathematically,

based on strength of covariance (in combination with other bands) with TOR EC for selection in quantitative prediction. Similar approaches based on covariance analysis methods have reported acetyl, aromatic, and phenol structures (Bornemann et al., 2008) or alkane C-H (Rosa Arranz et al., 2013) to predict the abundance of recalcitrant or black carbon in soils. For predicting TOR EC in atmospheric samples, it is possible that our calibration models not only use graphitic structure of EC, but rely on fragments of co-emitted OC, or artifact from the partitioning of total carbon between OC and EC by TOR. Based on

our analysis of FGs in Section 3.3.1, we cannot rule out at present time that some of these organic FG assignments may arise from spurious correlations. While surface FGs of soot particles sampled at source have been detected by FT-IR (Cain et al., 2010), we consider that their potential mass contribution is insignificant with respect to the overall mass of accompanying organic PM that possibility of extracting this surface FG contribution to the overall organic signal is unlikely. We anticipate that plausibility of these hypotheses can be further constrained in future studies.





There are additional bands which appear to be relevant for one spectra type but not the other for both TOR OC and EC (and also other analytes discussed in Section 3.3.1), and as described in the following section (3.3.3), more models can be constructed that includes a larger number of wavenumbers. It is unclear whether these additional FGs are superfluous, complementary, or spurious in their relation to groups discussed above, but the value of their absorbances were not ruled out in the model

selection process. The most reasonable interpretation of relevant FG lies somewhere between the extremes of the common FG and divergent FG hypothesis, and further investigation on this topic is also left for future work.

### 3.3.3 Interpretation of TOR OC and EC solutions from SPLSa, SPLSb, and full models

SPLSa and SPLSb solutions for TOR OC and EC calibration models developed with baseline corrected spectra use wavenumbers similar to EN-PLS which are described above. In both TOR OC and EC models, additional wavenumbers in the range

$2200$–$2500 \ \mathrm{cm}^{-1}$ and above $3300 \ \mathrm{cm}^{-1}$ are used by SPLSa and SPLSb solutions. The former set of wavenumbers may correspond to vibrational modes from isocyanates, nitriles, and phosphines, and usually have very low absorbance in ambient samples. This may explain why the most parsimonious solutions from EN-PLS do not use this range of wavenumbers. The wavenumbers above $3300 \ \mathrm{cm}^{-1}$ may correspond the absorption of alcohol and amine FGs, whose contribution to TOR OC and EC prediction is taken into account in other parts of the spectrum by the EN-PLS solution. SPLSa, SPLSb, and EN-PLS raw

spectra solutions for TOR OC and EC use the range near $1700 \ \mathrm{cm}^{-1}$ attributed to C=O absorption, and the PTFE C-F stretching region to account for removing the PTFE contribution to the overall signal.

The full spectra solutions include features that have been previously described in the EN-PLS solutions, though more specific interpretation is more difficult for lack of sparsity. In the baseline corrected solution we can see that for both TOR OC and EC high VIP scores correspond to the regions around $1700$, $3000$ and $3400 \ \mathrm{cm}^{-1}$ associated with C=O, C-H and O-H stretching,

respectively. For TOR EC, the high VIP scores in the region of C-H stretching are associated with negative coefficients, as in the EN-PLS solution. For the raw spectra models of TOR OC and EC, the PTFE C-F stretching region is also used for background subtraction. The most distinguishing features with highest VIP scores associated with analytes are the C=O stretching (near $1700 \ \mathrm{cm}^{-1}$) for the TOR OC, and benzene ring stretch (near $1500 \ \mathrm{cm}^{-1}$) for the TOR EC.

### 4 Conclusions

We evaluated four sparse methods in the construction of calibration models for four organic FGs and TOR OC and EC. Since the full wavenumber models already performed well in prediction, the best of the sparse models generally did not improve model performance, but conferred interpretation regarding the most relevant absorption bands required for prediction. In formulating sparse models, the direct one-norm penalty on regression coefficients by EN permitted better control of sparsity — i.e., stronger correlations between penalty and sparsity were observed, and more sparse solutions were ultimately obtained — than imposing

penalties on individual weight or direction vectors as formulated by SPLSa and SPLSb.

SPLS methods were less robust than EN and EN-PLS in extrapolating calibration models developed with laboratory standards for use in estimating FG abundances in ambient samples. For example, FG-OC estimated by the full wavenumber PLS





model using raw spectra had a slope and correlation of 0.97 and 0.93 in comparison to TOR OC, while the performance dropped as low as 1.69 (slope) and 0.77 (correlation) with the SPLS methods. The additional dimensionality reduction applied by EN-PLS led to better performance than EN using the same wavenumbers, and similar performance to the full wavenumber models was achieved while using only 1–6 % of original wavenumbers for each FG. As some samples are more sensitive to model

formulation and sparsity, such methods can possibly be used to identify cases in which laboratory standards do not reflect the types of bonds in ambient samples. When sparse methods were used to build calibration models using ambient samples, TOR OC and EC prediction metrics were insensitive to sparsity. All PLS-based models predicted reference values (not included during calibration) with less than 10% bias and correlation coefficients higher than 0.9.

In examining sparse calibration models for aCOH, cCOH, aCH, and CO, selected wavenumbers for FGs are consistent with

known absorption bands of their constituent bonds. FGs contributing to our capability for prediction for TOR OC are those which are commonly associated with organic PM, while for TOR EC, the main bond found in common between the raw and baseline corrected spectra are C-C stretch in ring-structured compounds. Wavenumbers used by raw and baseline corrected spectra appear to vary significantly, but can be (and have been) interpreted through different vibrational modes associated with a common set of FGs.

This first evaluation of sparse calibration methods using FT-IR spectra shows promise in conferring interpretation of associated molecular bonds to TOR OC and EC measurements. Sparse calibration models "localized" (in the statistical sense) by spectral features can be used to identify key FGs used for prediction of TOR measurements at various sites (e.g., Reggente et al., 2016), and aid construction of calibration sets suitable for prediction of individual or groups of samples. Furthermore, this type of analysis demonstrates the capability to provide interpretation of molecular bonding to other quantifiable metrics of

complex PM to which spectral features from FT-IR can be correlated.

## Appendix A: Notation

Tables A1 and A2 summarize notation used for matrices and vectors with their corresponding dimensions. Matrices in written in uppercase italic bold and vectors in lowercase italic bold. Vectors are column vectors by convention; row vectors are written as transposed vectors.

**Table A1.** Dimensions and indexing variables.

| Scalar variable | Description | Dummy index |
|:---:|:---|:---:|
| $N$ | number of samples | $i$ |
| $M$ | number of independent variables (wavenumbers) | $j$ |
| $K$ | number of latent variables used | $h, k$ |



**Table A2.** Arrays and dimensions.

| Array variable | Vector/scalar notation | Description |
|---|---|---|
| $\boldsymbol{X}$ | $[\boldsymbol{x}_i^T]$ | matrix of spectra ($N \times M$) |
| $\boldsymbol{Y}$ | $[\boldsymbol{y}_k]$ | matrix of dependent variables ($N \times 1$) |
| $\boldsymbol{B}$ | $[\boldsymbol{b}_k]$ | matrix of PLS coefficients ($M \times 1$) |
| $\boldsymbol{T}$ | $[\boldsymbol{t}_h]$ | matrix of X scores ($N \times K$) |
| $\boldsymbol{P}$ | $[\boldsymbol{p}_h]$ | matrix of X loadings ($M \times K$) |
| $\boldsymbol{E}_X$ | $[\boldsymbol{e}_{x,i}^T]$ | matrix of X residuals ($N \times M$) |
| $\boldsymbol{Q}$ | $[\boldsymbol{q}_h]$ | matrix of Y loadings ($1 \times K$) |
| $\boldsymbol{E}_Y$ | $[\boldsymbol{e}_{y,i}^T]$ | matrix of Y residuals ($N \times 1$) |
| $\boldsymbol{R}$ | $[\boldsymbol{r}_h]$ | matrix of X direction vectors ($M \times K$) |
| $\boldsymbol{W}$ | $[\boldsymbol{w}_h]$ | matrix of X weights ($M \times K$) |

## Appendix B: Model specification

The derivation, properties, and implementation of sparse methods used in this manuscript are described in detail by their respective authors: SPLSa (Lê Cao et al., 2008), SPLSb (Chun and Keles, 2010), EN (Zou and Hastie, 2005; Friedman et al., 2010), and EN-PLS (Fu et al., 2011). In this section, we briefly summarize the methods using consistent notation such that 1) their problem statements can be compared through their objective functions and constraints (formulated as penalties), and 2) how sparsity is controlled by their respective tuning parameters is apparent. An overview of methods and the parameters over which models are explored is provided in Table B1. For PLS-methods, the more general case for multivariate $\boldsymbol{Y}$ is introduced, and specific simplifications for univariate $\boldsymbol{y}$ is described where notable. For solving the PLS problem, we use the NIPALS algorithm in each case as the weight vectors derived from this algorithm can be used for calculating VIP scores.

## B1    Partial Least Squares (PLS)

A search for LVs can be framed as an optimization problem to maximize covariance between response and explanatory variables under a set of transformations (Burnham et al., 1996; Chun and Keles, 2010; Lee et al., 2011; Filzmoser et al., 2012; Liu, 2014). Writing the matrix product of the spectra and response variables as $\boldsymbol{Z} = \boldsymbol{X}^T\boldsymbol{Y}$, the transformations are introduced through the weight vector $\boldsymbol{w}$ for each $k$th LV:

$$\begin{aligned} \underset{\boldsymbol{w}_k}{\arg\max} \quad & \boldsymbol{w}_k^T \boldsymbol{Z}_k \boldsymbol{Z}_k^T \boldsymbol{w}_k \\ \text{s.t.} \quad & \|\boldsymbol{w}_k\|^2 = 1 \end{aligned} \tag{B1}$$

with a constraint to ensure that the weight vectors are normalized. In the non-linear iterative partial least squares (NI-PALS) algorithm (Wold, 1966; Martens, 1991), the weight vectors are calculated from the deflated (residual) matrix $\boldsymbol{Z}_k = \boldsymbol{X}_k^T\boldsymbol{Y}_k$ obtained from the $k$th iteration (Burnham et al., 1996; Chun and Keles, 2010; Lee et al., 2011) in which





$X_k = (I_N - T_{k-1}T_{k-1}^+)X_{k-1}$ and $Y_k = (I_N - T_{k-1}T_{k-1}^+)Y_{k-1}$. $I_N$ is the identity matrix of dimension $N \times N$, $T_{k-1} = [X_1 w_1, X_2 w_2, \ldots, X_{k-1} w_{k-1}]$ and $T^+$ is the Moore-Penrose inverse of $T$. $T_0 \equiv 0_{N \times K}$ such that $X_0 = X$ and $Y_0 = Y$ (Burnham et al., 1996; ter Braak and de Jong, 1998; Lee et al., 2011). The weight vectors correspond to eigenvectors of $Z_k Z_k^T$ (Höskuldsson, 1988; Rosipal and Krämer, 2006).

$w$ and $r$ introduced as column elements of $W$ and $R$, respectively, in Section 2.2 are related in concept and often referred to as loading weights, loadings, weights, and direction vectors interchangeably (e.g., Haaland and Thomas, 1988; Kvalheim and Karstang, 1989; Mevik and Wehrens, 2007; Lê Cao et al., 2008; Chun and Keles, 2010; Lee et al., 2011; Filzmoser et al., 2012). In this manuscript, we adopt the convention of referring to $w$ as (loading) weights and $r$ as direction vectors, respectively. Using the definition of deflated matrices, we can also write the relationship between the loading weights and

direction vectors as $X_k w_k = X r_k = t_k$, and $r_k = (I_M - RP^T)w_k$ where $I_M$ is the identity matrix of dimensions $M \times M$ (ter Braak and de Jong, 1998). The reader will note that $Z$ is proportional to the cross-covariance matrix between $X$ and $Y$, and the objective function (Equation B1) is in fact proportional to the inner product of the cross covariances between $Y_k$ and the transformed variables $t_k$ for each LV $k$.

       To solve for the underlying weights and direction vectors which satisfy these equations, we use the NIPALS algorithm

implemented in the `pls` library (Mevik and Wehrens, 2007) for the R programming language (R Core Team, 2014) in this work. Candidate models are generated by varying $K = \{1, 2, \ldots, 120\}$, from which one is selected by penalizing model variance over an ensemble of scaling factors and combining them through consensus scoring (Takahama and Dillner, 2015).

## B2   Elastic Net (EN) regularization

EN regularization is not a variant of PLS but solves for regression coefficients in Equation 1 without using LVs. The objective

function to be minimized is similar to the residual sum-of-squares (RSS) used in ordinary least squares regression, but with additional constraints imposed on the regression vector (Zou and Hastie, 2005):

$$\arg\min_b \|y - Xb\|_2^2 + \lambda_1 \|b\|_1 + \lambda_2 \|b\|_2^2 . \tag{B2}$$

The first penalty corresponds to that used by LASSO regression (Tibshirani, 1996) and imposes sparseness constraints, while the second penalty corresponds to that used by ridge regression (Hoerl and Kennard, 1970) or standard Tikhonov regularization

(Tikhonov and Arsenin, 1977) and imposes restrictions on the overall size of the regression vector. Combining the two penalties imposes sparsity constraints but retains a grouping effect which enables selection of co-varying variables together (Zou and Hastie, 2005). Without the second penalty, the LASSO penalty alone often selects only one of the covariates at random (leading to potential loss of relevant variables), and permits at most $N$ variables to be retained (Zou and Hastie, 2005), which is not always desirable when the inverse problem is underdetermined. In practice, the parameter space for EN is not formulated in

terms of $\lambda_1$ and $\lambda_2$ but by the overall penalty $\lambda = \lambda_1 + \lambda_2/2$ and the fractional contribution of the first penalty $\alpha = \lambda_1/\lambda$ such that the overall penalty to RSS is written as $p_{\lambda,\alpha}(b) = \lambda \left[ \alpha \|b\|_1 + (1/2)(1-\alpha)\|b\|_2^2 \right]$. $\alpha = 0$ corresponds to LASSO regression and $\alpha = 1$ corresponds to ridge regression. Though $\alpha$ can be treated as a free parameter spanning values between 0 and 1, we fix the value at $\alpha = 0.5$ to mix the penalties for our EN regression solution (Friedman et al., 2010; Hastie and Qian,




2014). As part of the algorithm, a scaling correction is applied a posteriori to the "naïve" regression coefficients specified by Equation B2 to correct for biases introduced through application of both the ridge and LASSO regularization procedures in obtaining the regression vector (Zou and Hastie, 2005).

We perform EN regression using the `glmnet` library (Friedman et al., 2010) in R. We generate 100 models (regression coefficients) for various values of $\lambda$ spanning a specified range for each variable. As an upper bound, $\lambda$ for which all regression coefficients become zero is selected, and $10^{-3}$ times this upper value is selected as a lower bound.

## B3  Sparse PLS by penalty on the weight vector (SPLSa)

A sparse PLS formulation by Lê Cao et al. (2008) is inspired by the sparse principal component analysis (PCA) of Shen and Huang (2008), and based on a singular value decomposition (SVD) estimate of the direction vector with sparsity imposed by a LASSO penalty function. For each $k$th LV, $\boldsymbol{Z}_k$ is decomposed by its rank-one approximation from SVD, and modified singular vectors which additionally satisfy the sparsity constraints are sought by iterative calculation:

$$\underset{\boldsymbol{w}_k, \boldsymbol{v}_k}{\arg\min} \; \|\boldsymbol{Z}_k - d_k \boldsymbol{w}_k \boldsymbol{v}_k^T\|_F^2 + g_{\lambda_1}(\boldsymbol{w}_k) + g_{\lambda_2}(\boldsymbol{v}_k) \,.$$

$\boldsymbol{w}_k$ and $\boldsymbol{v}_k$ are the left and right singular vectors of $\boldsymbol{Z}_k$, respectively; as eigenvectors of $\boldsymbol{Z}_k \boldsymbol{Z}_k^T$ and $\boldsymbol{Z}_k^T \boldsymbol{Z}_k$ they are also equivalent to loading weights for $\boldsymbol{X}$ and $\boldsymbol{Y}$ (Lê Cao et al., 2008). $d_k$ is the corresponding singular value from the SVD. $\|\cdot\|_F$ is the Frobenius norm, $\boldsymbol{Z}_k$ is the deflated matrix after subtraction of the first $k-1$ components ($\boldsymbol{Z}_k = \boldsymbol{Z} - \boldsymbol{W}_{k-1}\boldsymbol{D}_{k-1}\boldsymbol{V}_{k-1}^T$). While the expression above is written generally to accommodate a multivariate $\boldsymbol{Y}$ scenario, $\boldsymbol{y}$ is a $N \times 1$ column vector in our specification. Therefore, $\boldsymbol{w}_k$ is the $k$th weight vector in PLS formulation (hence the labeling of this vector as $\boldsymbol{w}$); $\boldsymbol{v}_k \equiv 1$ for all $k$ and the second penalty term on this vector is not necessary. Updated weight vectors are obtained from a soft thresholding operator (Shen and Huang, 2008; Francis Bach and Obozinski, 2011; Mazumder et al., 2011; Mehmood et al., 2012) applied at each iteration (Lê Cao et al., 2008): $\tilde{\boldsymbol{w}}_k = g_{\lambda_1}(\hat{\boldsymbol{w}}_k) = \text{sign}(\hat{\boldsymbol{w}}_k) \cdot \max\{0, |\hat{\boldsymbol{w}}_k| - \lambda_1\}$, and corresponds to the LASSO penalty function (Hastie et al., 2009).

This algorithm is implemented in the `mixOmics` package (Dejean et al., 2014) in R. The soft threshold ($\lambda_1$) is selected accordingly by the magnitude of the $j$-th highest loading weight, which we write as $n_X$, when sorted by decreasing magnitude. Therefore, the threshold or tuning parameter is specified by the number of variables to retain in each LV according to user input. We select values of $n_X = \{10, 20, 30, 50, 70, 100, 200, 300, 500, 1000, 1500, 2000, 2500\}$; as permitted by the maximum number of wavenumbers $M$ in the input spectra. We choose $K = \{1, 2, \ldots, 35\}$. Because the wavenumbers for loading weights falling below the $j$th-ranked value are different for each LV, the specification by number of variables does not necessarily lead to the same number of non-zero coefficients in the regression vector. The number of non-zero coefficients is calculated from the solutions presented in Section 2.

## B4  Sparse PLS by penalty on a surrogate vector (SPLSb)

Another sparse PLS algorithm, introduced by Chun and Keles (2010), is based on the sparse principal component analysis (PCA) by Zou et al. (2006) combined with the EN penalty of Zou and Hastie (2005). Rather than imposing a sparsity constraint





on the weight or direction vector, a higher degree of sparsity is targeted by placing a constraint on a surrogate $\boldsymbol{c}$ of the direction vector:

$$\arg\min_{\boldsymbol{r}_k, \boldsymbol{c}_k} \quad -\kappa \boldsymbol{r}_k^T \boldsymbol{Z}\boldsymbol{Z}^T \boldsymbol{r}_k + (1-\kappa)(\boldsymbol{c}_k - \boldsymbol{r}_k)^T \boldsymbol{Z}\boldsymbol{Z}^T (\boldsymbol{c}_k - \boldsymbol{r}_k) + \lambda_1 \|\boldsymbol{c}_k\|_1 + \lambda_2 \|\boldsymbol{c}_k\|_2^2$$
$$\text{s.t.} \quad \|\boldsymbol{r}_k\|_2 = 1 \ .$$

The first term maximizes covariance (by minimizing its negative value) as in the original PLS problem (Equation B1); the second term keeps the surrogate vector $\boldsymbol{c}$ in close alignment with the direction vector $\boldsymbol{r}$, with $\kappa$ controlling the tradeoff between the two terms. The penalty, similar to that of EN, balances sparsity with preventing a potential singularity in the inversion of $\boldsymbol{Z}\boldsymbol{Z}^T$. The solution in the multivariate case is sought by fixing $\boldsymbol{w}$ or $\boldsymbol{c}$ and solving for the other vector in an alternate fashion (Chun and Keles, 2010). In practice, $\lambda_2$ is chosen to be large for the estimator to be obtained by soft thresholding; for univariate $\boldsymbol{y}$ the solution does not depend on $\kappa$ so it is fixed to a value of 1/2 (Chun and Keles, 2010; Filzmoser et al., 2012). Therefore, the key dependence can be reduced from four to two parameters, $K$ and $\lambda_1$. For the univariate case, the solution for $\mathbf{c}_k$ is related to the direction vector $\hat{\mathbf{c}}_k = \text{sign}(\hat{\boldsymbol{r}}_k) \cdot \max\{0, \hat{\boldsymbol{r}}_k - \lambda_1/2\}$ (Chun and Keles, 2010). As with SPLSa, the penalty with respect to $\lambda_1$ is reformulated as a soft thresholding operator containing the bounded parameter $\eta$ ($0 \leq \eta \leq 1$) to be applied at each iteration (Chun and Keles, 2010; Filzmoser et al., 2012): $\tilde{\boldsymbol{c}}_k = \text{sign}(\hat{\boldsymbol{r}}_k) \cdot \max\left\{0, |\hat{\boldsymbol{r}}_k| - \eta \max_{1 \leq j \leq M} |\hat{r}_{j,k}|\right\}$. Rather than thresholding on the $j$th value of a vector as described for SPLSa, components of the direction vectors below a fraction $\eta$ of the magnitude of the largest component are set to zero. The solutions obtained for $\hat{\boldsymbol{r}}$ are only used for variable selection, and ordinary PLS is used on the selected wavenumbers to obtain the final sparse solution (Chun and Keles, 2010).

We use the implementation in the `spls` library (Chung et al., 2013) in R. We select values of the thresholding parameter as $\eta = \{0.1, 0.2, \ldots, 0.9, 0.92, 0.95, 0.98\}$ and the number of components over the same domain for SPLSa: $K = \{1, 2, \ldots, 35\}$. Internally, the final fitting is performed via the `pls` library, and for this we also specify use of the NIPALS algorithm.

## B5   EN combined with PLS (EN-PLS)

Fu et al. (2011) introduced a two-step strategy by which EN regression is used for preliminary variable selection, and a final calibration model is developed by projection onto LVs with PLS regression. In the proposition by Fu et al. (2011), additional backward variable selection on groups of contiguous wavenumbers is incorporated into the second stage of the procedure; models are iteratively constructed with diminishing subsets of wavenumbers until the apparent performance with respect to a defined criterion (e.g., minimum RMSECV) no longer improves. As the definition of this criterion may not be consistent with the criteria by which we evaluate and select the final model for this work (e.g., parsimony, comparison with ambient reference measurements), we forgo this additional wavenumber selection step and use the wavenumbers selected by EN regression directly.

We have implemented this method in R in combination with the `pls` library and NIPALS algorithm as described above. Variable selection with PLS is applied to calibration models developed with TOR measurements. PLS regression using non-zero wavenumbers from EN regression is also performed with laboratory standards for FG estimation, but the additional





variable selection is not used since the best model for extrapolation to ambient samples may not necessarily be selected by lower $RMSE_d$ (Takahama and Dillner, 2015).

**Table B1.** Summary of sparse methods and parameters.

| Method | Parameter | Values |
|--------|-----------|--------|
| EN | sparsity | $\alpha = 0.5, \lambda = \{10^{-3}\lambda_{\max}, ..., \lambda_{\max}\}$ |
| SPLSa | sparsity | $\eta = \{0.1, 0.2, \ldots, 0.9, 0.92, 0.95, 0.98\}$ |
| | LVs | $K = \{1, 2, \ldots, 35\}$ |
| SPLSb | sparsity | $n_X = \{10, 20, 30, 50, 70, 100, 200, 300, 500, 1000, 1500, (2000, 2500)*\}$ |
| | LVs | $K = \{1, 2, \ldots, 35\}$ |
| EN-PLS | sparsity | fixed by EN |
| | LVs | $K = \{1, 2, \ldots, 35\}$ |

∗Values in parentheses correspond to values explored only in raw spectra models as baseline corrected models cannot exceed the value of 1563 (the total number of wavenumbers).

## Appendix C: Additional remarks on model selection

### C1 Metrics for model selection

The weighting of the bias and variance can be formulated in various functional forms with tuning parameters, and the final model chosen by consensus scoring (e.g., Sum of Ranking Differences; Héberger, 2010; Héberger and Kollár-Hunek, 2011; Kollár-Hunek and Héberger, 2013) when the most suitable value for the parameter is not known a priori (Kalivas et al., 2015; Takahama and Dillner, 2015). The solutions for the full wavenumber PLS solutions are obtained using this method. Given the additional information required (RMSE values for each CV fold at every definition of sparsity and LVs), we forgo applying

this formal procedure for the sparse PLS methods. Model selection according to the pRMSECV criterion will yield the number of LVs as using metric $M2_k(\lambda)$ (Takahama and Dillner, 2015) and setting the penalty parameter to its characteristic value, $\lambda = \lambda^*$, which corresponds to the extreme limit of penalties considered in the evaluation of this metric.

### C2 Biases in RMSECV

Variable selection and model parameter estimation is performed entirely within the calibration set for all methods. The two

objectives are combined in EN and SPLSa by their respective soft thresholding penalties, but are separated into two steps by SPLSb and EN-PLS. SPLSb correctly applies both variable selection and parameter estimation prior to estimation of error and against the validation (sub)set, while our current implementation of EN-PLS method performs variable selection (via EN) prior to the CV and RMSECV estimation procedure of the second step. In the latter scenario, the reported RMSECV can be underestimated (Hastie et al., 2009; Lee et al., 2011) as the constructed model the validation samples have already informed

construction of the model (for wavenumber reduction), albeit according to different set of selection criteria. We note that the





implication of this bias is the lack of comparability with RMSECV values estimated with other algorithms. While this is a potential area for further improvement, our model selection criteria use relative RMSECV estimates; we expect that this bias will not have a major influence on model selection and no affect on reported evaluation against test set samples.

*Acknowledgements.* The authors acknowledge funding from the Swiss National Science Foundation (200021_143298) and the IMPROVE
5    program (National Park Service cooperative agreement P11AC91045). We also thank A. Weakley for helpful discussions.



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



# Tables

**Table 1.** Number of wavenumbers and LVs selected for final models.

| Spectra type | Method | number of NZVs (percent of full), number of LVs | | | |
| --- | --- | --- | --- | --- | --- |
| | | aCOH | cCOH | aCH | CO |
| Raw | Full | 2784 (100%), 13 | 2784 (100%), 17 | 2784 (100%), 10 | 2784 (100%), 18 |
| Raw | SPLSa | 549 (20%), 13 | 1022 (37%), 4 | 1837 (66%), 12 | 178 (6%), 6 |
| Raw | SPLSb | 107 (4%), 10 | 237 (9%), 7 | 2029 (73%), 15 | 1464 (53%), 4 |
| Raw | EN | 94 (3%), none | 102 (4%), none | 40 (1%), none | 54 (2%), none |
| Raw | EN-PLS | 94 (3%), 12 | 102 (4%), 7 | 40 (1%), 7 | 54 (2%), 7 |
| Baseline corrected | Full | 1563 (100%), 22 | 1563 (100%), 9 | 1563 (100%), 18 | 1563 (100%), 9 |
| Baseline corrected | SPLSa | 172 (11%), 6 | 100 (6%), 1 | 1451 (93%), 9 | 40 (3%), 4 |
| Baseline corrected | SPLSb | 483 (31%), 31 | 236 (15%), 9 | 147 (9%), 9 | 248 (16%), 9 |
| Baseline corrected | EN | 99 (6%), none | 47 (3%), none | 91 (6%), none | 79 (5%), none |
| Baseline corrected | EN-PLS | 99 (6%), 19 | 47 (3%), 15 | 91 (6%), 13 | 79 (5%), 16 |

| Spectra type | Method | number of NZVs (percent of full), number of LVs | |
| --- | --- | --- | --- |
| | | OC | EC |
| Raw | Full | 2784 (100%), 48 | 2784 (100%), 28 |
| Raw | SPLSa | 2784 (100%), 8 | 2565 (92%), 15 |
| Raw | SPLSb | 629 (23%), 8 | 160 (6%), 14 |
| Raw | EN | 194 (7%), none | 113 (4%), none |
| Raw | EN-PLS | 194 (7%), 7 | 113 (4%), 8 |
| Baseline corrected | Full | 1563 (100%), 15 | 1563 (100%), 33 |
| Baseline corrected | SPLSa | 895 (57%), 12 | 828 (53%), 15 |
| Baseline corrected | SPLSb | 331 (21%), 7 | 1544 (99%), 12 |
| Baseline corrected | EN | 124 (8%), none | 143 (9%), none |
| Baseline corrected | EN-PLS | 124 (8%), 8 | 143 (9%), 9 |



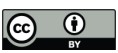

**Table 2.** Summary of FG and associated vibrational modes with positive regression coefficients used in the prediction of TOR OC and TOR EC by EN-PLS method. The FGs that are common to every solution (both raw and baseline corrected solution for TOR OC and EC) are reported first, followed by non-oxidized FGs, oxygenated FGs, and nitrogenated FGs; the order is not indicative of abundance.

| Functional groups | OC | | EC | |
| --- | --- | --- | --- | --- |
| | Raw spectra | Baseline corrected spectra | Raw spectra | Baseline corrected spectra |
| Aromatic | Ring stretch in aromatic compounds<br>Substituted benzene ring overtones†<br>Conjugation with C=O Naphtalenes in plane ring deformation | Conjugation with C=O<br>C=C stretch†<br>Substituted benzene ring overtones† | Benzene ring stretch | Benzene ring stretch<br>Conjugation with C=O<br>Substituted benzene ring overtones† |
| Amides | N-C=O bend<br>C=O out of plane bend<br>N-H bend | N-H bend | N-H bend | N-H bend |
| Esters | C=O stretch | C=O stretch | C-O-C antisymm. stretch | C=O stretch |
| Alkanes | CH₂ bend | C-H stretch | | C-H stretch |
| Alkenes | Conjugation with C=O | C=C stretch†<br>Conjugation with C=O | | Conjugation with C=O<br>Alkene C=C† |
| Carboxyl | C=O stretch<br>O-C=O bend | C=O stretch<br>OH stretch | | C=O stretch |
| Ketones | C=O stretch | C=O stretch | | C=O stretch |
| Aldehydes | C=O stretch<br>C-C-CHO bend | C=O stretch | | C=O stretch |
| Ethers | | | C-O stretch in aryl ethers | |
| Alcohol | C-O-H bend<br>Ar-OH out of plane deformation | OH stretch | | |
| Amines | | N-H bend | C-N stretch in aromatic amines | N-H bend |
| Nitro compounds | NO₂ deformation<br>NO₂ antisymm. stretch | | NO₂ antisymm. stretch | |

† weak bands



# Figures







**Figure 1.** Estimated RMSECVs for a range of models using different subset of wavenumbers is shown by shaded regions. Results are shown for models using raw spectra. For panels in the first two columns (SPLSa and SPLSb), the shaded regions extend from the minimum RMSECV to pRMSECV solutions for each sparsity penalization parameter. For the last column of panels (EN/EN–PLS), the shaded region extends from the minimum RMSECV to one standard error above for each value of the penalization parameter in EN estimates. Circles correspond to models selected for this work. For EN/EN-PLS panels, red circles correspond to the EN solution, and blue circles correspond to the selected solutions for EN-PLS. The RMSECVs for EN-PLS are underestimated (Section C) and may not be amenable for direct comparisons with other methods.





**Figure 2.** Same information as Figure 1 is shown for models using baseline corrected spectra.





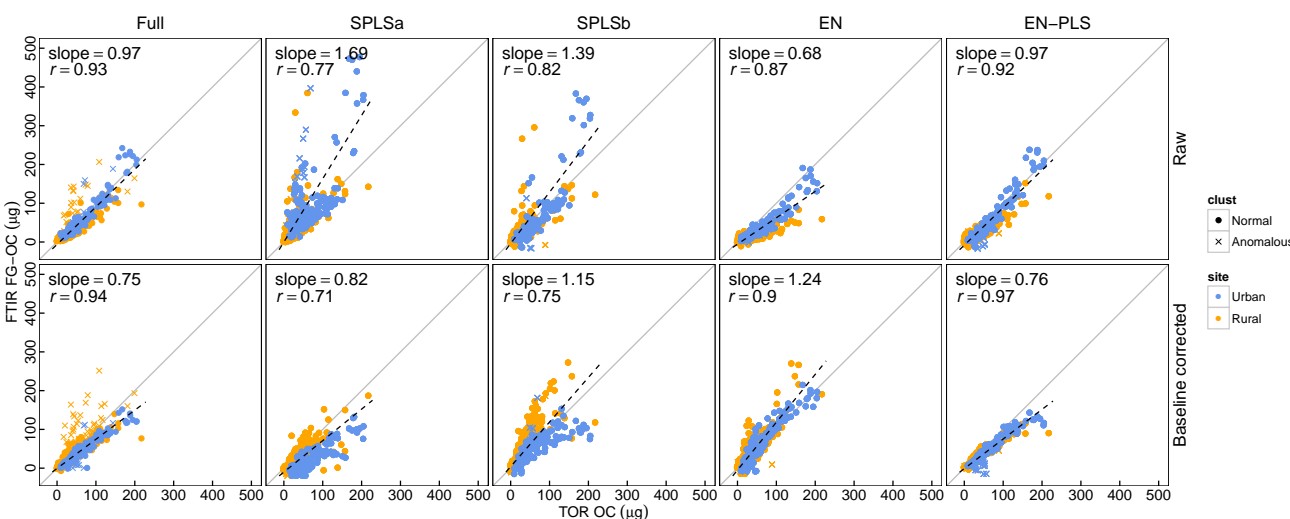

**Figure 3.** Predictions vs. reference for ambient samples (OC from sum of FG).





**Figure 4.** Comparison statistics of full and reduced wavenumber solutions show sensitivity of model predictions to sparsity. OC and EC correspond to predictions from direct calibration to TOR measurements rather than summing FGs.





**Figure 5.** Predictions vs. reference for ambient samples (direct calibration).



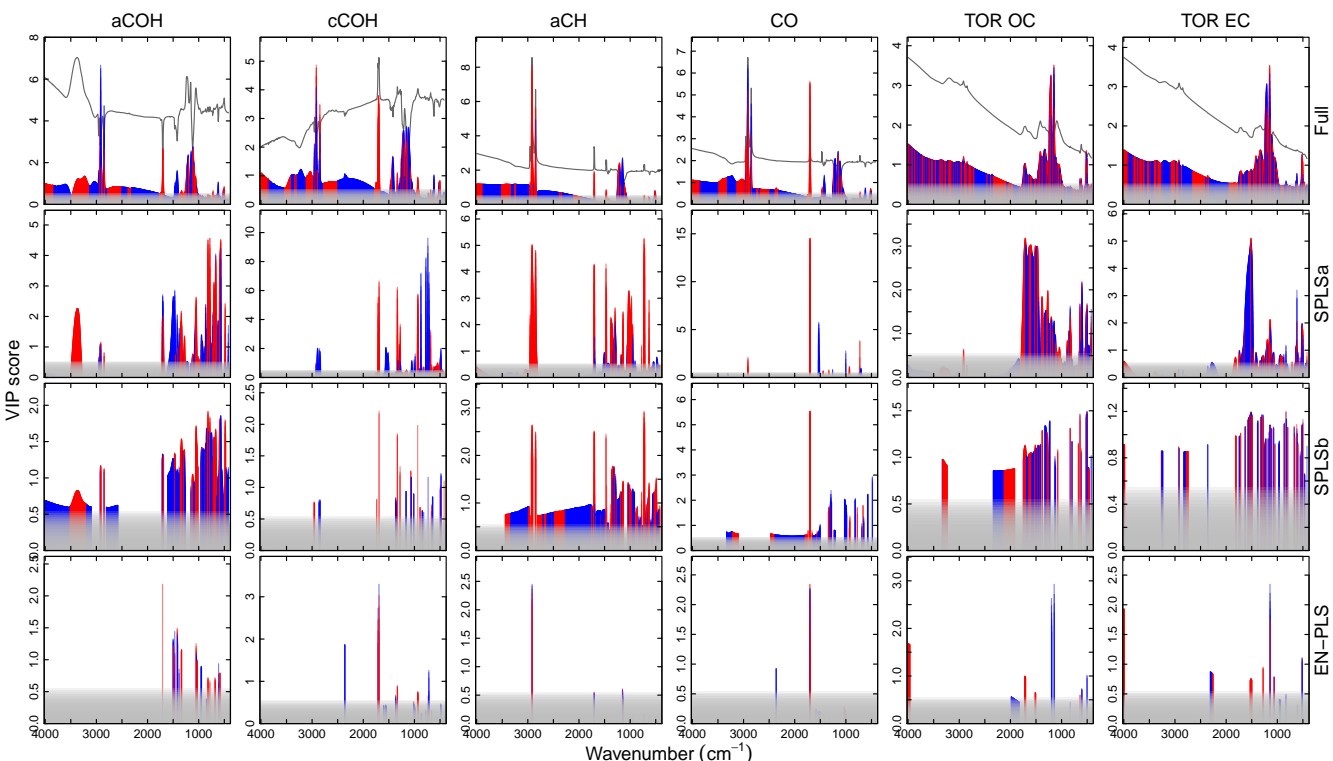

**Figure 6.** VIP for solutions using raw spectra. Dark gray lines in the "full" solution panels correspond to the first loading weights, and the vertical bars in every panel extend from zero to VIP scores. Red points accompanying vertical bars indicate wavenumbers for which regression coefficients are positive and blue points indiciate wavenumbers for which coefficients are negative. Regions up to VIP scores of 0.5 are shaded to indicate VIP scores not considered for our interpretation (Section 3.3).



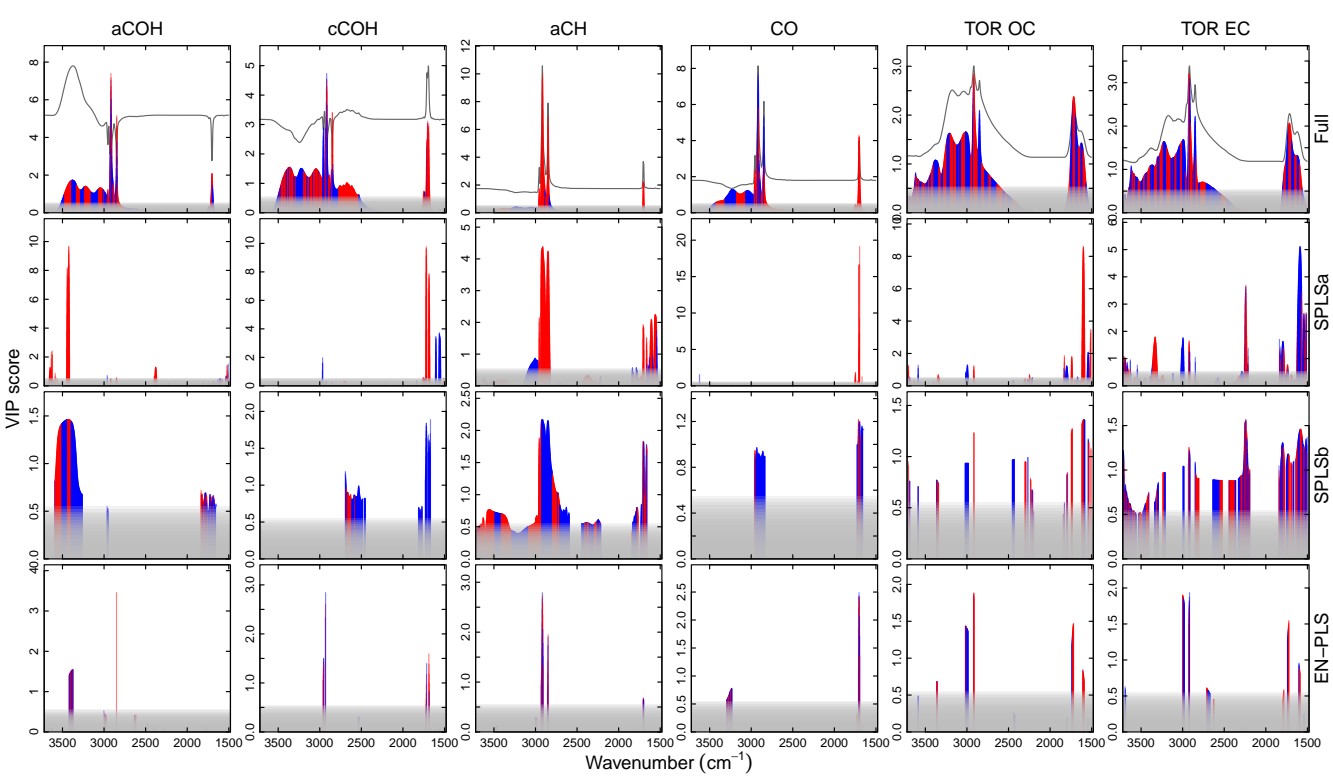

**Figure 7.** VIP for solutions using baseline corrected spectra. Lines and colors are as indicated for Figure 6.







**Figure 8.** Example group of baseline corrected spectra shown with VIP scores above 0.5 overlayed. The VIP is derived from all calibration spectra consisting of laboratory standards (same as those shown in Figure 7). The column of panels on right show sensitivity of predictions for each FG for this subset of samples.