# Peer review of "Supporting Information for "Analysis of functional groups in atmospheric aerosols by infrared spectroscopy: sparse methods for statistical selection of relevant absorption bands""

_Atmospheric Measurement Techniques, 2016_

## Referee Comment (RC1) · Anonymous Referee #1 · 16 Mar 2016

In this paper the authors present a very detailed and thorough anlysis of sparse methods for selection of relevant absorption bands in IR. The paper is well presented and the analysis is very detailed. I only have a few general comments regarding the general accuracy of methods according to reliability of the sampled data.

Page 2 line 24 onwards. Is the justification for sparse methods, in this sense, more with regards the end-point accuracy or prohibitive cost of other methods?

In section 2.1.1 the laboratory and ambient samples are discussed. It would be nice to see a brief consideration of instrument variability and how this might impact of the

performance of your evaluation. For example, is there an underlying assumption that there is no 'drift' in instrument response function across all datasets used? I wonder whether this might lead to the variability discussed in section 3.2 onwards? Perhaps I have misunderstood this, but is agreement between instruments proven?.

Towards the end of page 6 the authors discuss the rationale for choice of variable selection method. In section 2.3.1, how is the test fair in the sense of being able to cover a broad range of functionality expected in amboent samples? In other words, would it be possible to visualise where your test and training set 'sit' on a general 2D basis sets or even simple Ven Krevelen diagrams? Does this even matter?

On this point, there seems to be complicating factors in sample collection from ambient campaigns that might degrade the performance of any 'tuned' method. For example, could it be that whether any semi-volatile loss, for example, occured during sample preparation might lead to a functional group dependency of variable selection?. It might be prudent to assess such effects with a consideration of other factors such as RH, Temp, provenance of sample. Based on the high mass loadings displayed in figure 3, this might not yet be apparent but its worthwhile considering. I guess I'm wondering whether, aside from variable selection refining a fixed instrument response, the algorithms are also picking up 'noise' from external factors.

Minor comments:

Page 8, line 18: I think an 'if' is missing after 'examining' I would recommend expanding acronyms in all figure captions for ease of reading.

---

## Referee Comment (RC2) · Anonymous Referee #2 · 21 Mar 2016

In this well-structured and very detailed paper, four sparse algorithms (SPLSa, SPLSb, EN and EN-PLS) were considered and evaluated for selecting relevant mid-infrared absorption bands in the calibration model building process. Using FTIR spectra obtained in transmission mode, two types of sparse calibration models were constructed for predicting/interpreting: (1) abundances of four organic functional groups (alcohol COH, carboxylic COH, alkane CH and carbonyl CO) and (2) TOR OC and EC concentrations in ambient aerosol samples. The paper has also presented a thorough analysis of the constructed models. I find the paper suitable for publication in AMT and only have very few minor comments:

Page 3, line 75 onwards: Samples were taken from seven sites in the IMPROVE network. Did the investigators set out any criteria for selecting these sites for the study? It might be a good idea to provide a brief description of the sites to gauge their representativeness of the network. Throughout the paper, nothing was mentioned about these sites except that they were comprised of rural and urban sites (from Figures 3 and 4).

Page 4, line 100: The 250 laboratory standards used were mixtures of seven compound types. What are these compounds? If this has been described elsewhere, citing the relevant literature would suffice.

Page 40, Figure 3 legend: What do the investigators mean by anomalous clusters?

PM: PM10 or PM2.5?

Grammatical/typographical corrections:

Page 38, Figure 1 caption: Should this be Appendix C instead of Section C?

Please go over the manuscript again and proofread it.

---

## Author Comment (AC1) · 14 Jun 2016

**Response to Reviewer #1**

In this paper the authors present a very detailed and thorough anlysis of sparse methods for selection of relevant absorption bands in IR. The paper is well presented and the analysis is very detailed. I only have a few general comments regarding the general accuracy of methods according to reliability of the sampled data.

We thank the reviewer for the positive comments.

1. Page 2 line 24 onwards. Is the justification for sparse methods, in this sense, more with regards the end-point accuracy or prohibitive cost of other methods?

We would like to use this opportunity to discuss our separate motivations for 1) using sparse methods, and 2) FT-IR more generally.

The motivation for implementing sparse methods is to improve end-point accuracy and interpretation of aerosol composition by FT-IR. Uninformative variables (i.e., wavenumbers) can add variance, or in some instances, bias to the predictions. In this work, the advantage with respect to end-point accuracy is not very clear, as the original full wavenumber models were already quite accurate. However, elimination of uninformative variables helps drive our interpretation regarding the most important absorption bands necessary for predicting TOR OC and EC, as well as FG, which allows us to make associations between carbonaceous aerosol and their molecular structure.

Our focus on developing quantiative algorithms FT-IR more generally is motivated for its potential to provide 1) a reduction in cost of aerosol monitoring, and 2) the additional chemical resolution (via functional groups) not permitted by other instruments. FT-IR analysis of each PTFE sample requires only a few minutes and can be integrated into existing analysis toolchains without the need for a separate sampling line for aerosol collection on quartz fiber filters (specifically for thermal optical analysis). In addition, absorption by vibrational modes observed by FT-IR can inform us of aerosol molecular structure not provided by organic carbon analysis, or even common mass spectrometry techniques. We anticipate that FG information harvested from such measurements can be made useful for systematic model-measurement evaluation (e.g., Ruggeri et al., 2016).

As the first point is the focus of (Ruthenburg et al., 2014; Dillner and Takahama, 2015a,b; Reggente et al., 2016), we include this addition to the Abstract:

"Various vibrational modes present in molecular mixtures of laboratory and atmospheric aerosols give rise to complex Fourier Transform Infrared (FT-IR) absorption spectra. Such spectra can be chemically-informative, but often require sophisticated algorithms for quantitative characterization of aerosol composition. Naïve statistical calibration models developed for quantification employ the full suite of wavenumbers available from a set of spectra, leading to loss of mechanistic interpretation between chemical composition and the resulting changes in absorption patterns that underpin their predictive capability. Using sparse representations of the same set of spectra, alternative calibration models can be built in which only a select group of absorption bands are used to make quantitative prediction of various aerosol properties. Such models are desirable as they allow us to make association of predicted properties with their underlying molecular structure."

2. In section 2.1.1 the laboratory and ambient samples are discussed. It would be nice to see a brief consideration of instrument variability and how this might impact of the performance of your evaluation. For example, is there an underlying assumption that there is no 'drift' in instrument response function across all datasets used? I wonder whether this might lead to the variability discussed in section 3.2 onwards? Perhaps I have misunderstood this, but is agreement between instruments proven?

The agreement between instruments for this dataset has indeed been established by Dillner and Takahama (2015a,b). We have included this statement in the Introduction section (1):

"We remark that while the latter two carbonaceous substances [TOR OC and EC] are presumably are composed of a complex combination of molecules, statistical calibration models using FT-IR spectra have been demonstrated to accurately predict TOR-equivalent OC and EC within measurement precision of collocated samples reported by the TOR method (Dillner and Takahama, 2015a,b)."

and in Methods section (2.3.1):

"Full wavenumber calibration models for TOR OC and EC developed using this protocol have demonstrated ability to capture overall variations in concentrations which span a wide range of PM composition and environmental conditions (Dillner and Takahama, 2015a,b)."

Regarding the specific issue of instrument drift, we do not attribute discrepancies between predictions to changes in instrument response for this measurements follow work. The TOR established protocols QA/QCfor (http://vista.cira.colostate.edu/improve/data/QA\_QC/qa\_qc\_Branch.htm) inwhich samples analyzed during periods of excessive drift are reanalyzed. We are preparing separate documentation of performance evaluation for FT-IR, but repeated analysis of check standard samples indicate less than 2–5% drift in signal response over 1.5 years.

3. Towards the end of page 6 the authors discuss the rationale for choice of variable selection method. In section 2.3.1, how is the test fair in the sense of being able to cover a broad range of functionality expected in amb[i]ent samples? In other words, would it be possible to visualise where your test and training set 'sit' on a general 2D basis sets or even simple Ven Krevelen diagrams? Does this even matter?

We thank the reviewer for this question.

The range in functionality expected in ambient samples does guide our choice of calibration set samples and the final variables selected (though not the variable selection procedure, as this is statistically determined).

Apart from the chamber studies of Chhabra et al. (2011), we have not evaluated the O/C and H/C ratio characterized by FT-IR measurements in detail. This is an area of current work. However, the range in OM/OC ratio (which can be closely associated with O/C; Aiken et al., 2008) over the IMPROVE 2011 samples is described and presented by Ruthenburg et al. (2014). Selection of laboratory standards used for prediction of FG is not selected on the basis of OM/OC, however, but on the multidimensional space of the FGs such that overlapping absorption bands can be taken into account in the model development. One of the reasons for the larger range in predicted FG abundance among models is due to the fact that the laboratory standards and ambient samples are different, and constraining the model development and selection under these circumstances is an area

of current research.

For TOR OC analysis, Dillner and Takahama (2015a) explored the influence of aerosol composition on calibration set design. Aerosol composition was parameterized as OM/OC ratio (derived from estimates of Ruthenburg et al., 2014) and ammonium/OC ratio, and we found, not unexpectedly, that capability for prediction was dependent on the range of concentration and similarity composition between the calibration (which comprises training and validation) and test set samples. Similar conclusions for EC are drawn (Dillner and Takahama, 2015b). Therefore, the calibration and test set samples are selected to span the same composition space (e.g., as might be represented by van Krevlin diagram) by design. The variable (wavenumbers) selected for TOR OC and EC are specific to the composition of PM sampled in these seven sites for 2011, though Reggente et al. (2016) suggests the analysis can be extended for the same sites to 2013. More importantly, we feel that the main contribution of this manuscript is to introduce this technique and open up a new avenue of research, in which we can associate vibrational modes and molecular structure with aerosol of different composition (e.g., targeting specific regions in the van Krevlin space) characterized by collocated measurements; not limited to TOR. Such work is currently underway in our group.

We have included the following statement in the Methods section (2.1.1):

"The OM/OC ratio estimated in ambient samples span a range of 1.46 and 2.01 between the 10th and 90th percentiles, with a median ratio of 1.69 (Ruthenburg et al., 2014)."

and the following statement in the Methods section (2.3.1):

"The calibration and test sets are constructed identically to the most accurate class of full wavenumber models described previously (Ruthenburg et al., 2014; Takahama and Dillner, 2015; Dillner and Takahama, 2015a,b)."

and the following statement in the Conclusion section (4):

"Sparse calibration models "localized" (in the statistical sense) by spectral features or external variables can be used to identify key FGs used for prediction of TOR measurements at various sites (e.g., Reggente et al., 2016), and aid construction of calibration sets suitable for prediction of individual or groups of samples (e.g., stratified by environmental conditions or chemical composition). This work provides a demonstration of how molecular structure can be associated with other quantifiable metrics of complex PM to which spectral features from FT-IR can be correlated."

4. On this point, there seems to be complicating factors in sample collection from ambient campaigns that might degrade the performance of any 'tuned' method. For example, could it be that whether any semi-volatile loss, for example, occured during sample preparation might lead to a functional group dependency of variable selection? It might be prudent to assess such effects with a consideration of other factors such as RH, Temp, provenance of sample. Based on the high mass loadings displayed in figure 3, this might not yet be apparent but its worthwhile considering. I guess I'm wondering whether, aside from variable selection refining a fixed instrument response, the algorithms are also picking up 'noise' from external factors.

We thank the reviewer for raising this important point. Investigation of artifacts, and particularly semi-volatile species, is an active area of investigation for us and will be addressed in a series of future studies. However, for the purposes of this manuscript, we note that Ruthenburg et al. (2014); Takahama and Dillner (2015); Dillner and Takahama (2015a,b) have shown that a single set of calibration models can make accurate predictions for each of the response variables studied. Predicted concentrations of individual FGs have not been independently validated, but have been evaluated to the extent possible using agreement with TOR OC and trends in OM/OC ratios.

Therefore, to a first order we believe that we have been able to capture the varying range in aerosol composition on these filters due to effects of RH, temperature, semivolatile losses with our base case (full wavenumber) models. As the TOR OC and EC estimates are calibrated to collocated measurements, there is a question whether sampling artifacts may be different. Quartz fiber filters have vapor adsorption artifacts, but have been corrected in a "mean" way for OC (Dillner and Takahama, 2015a).

We have revised Methods section (2.1.1):

"Monthly median values of OC loadings in blank samples are subtracted from ambient TOR OC loadings to account for the gas phase adsorption artifact by quartz fiber filters (Dillner and Takahama, 2015a)."

and Methods section (2.3.1):

"Full wavenumber calibration models for TOR OC and EC developed using this protocol have demonstrated ability to capture overall variations in concentrations which span a wide range of PM composition and environmental conditions (Dillner and Takahama, 2015a,b)."

5. [Minor comment] Page 8, line 18: I think an if is missing after examining I would recommend expanding acronyms in all figure captions for ease of reading.

We have rephrased this sentence to read:

"Examining sparse regression coefficients are informative for identifying important absorption bands, but interpretation can still be complicated by the compensation of interfering bands (Haaland and Thomas, 1988; Kvalheim et al., 2014)."

**References**

- Aiken, A. C., Decarlo, P. F., Kroll, J. H., Worsnop, D. R., Huffman, J. A., Docherty, K. S., Ulbrich, I. M., Mohr, C., Kimmel, J. R., Sueper, D., Sun, Y., Zhang, Q., Trimborn, A., Northway, M., Ziemann, P. J., Canagaratna, M. R., Onasch, T. B., Alfarra, M. R., Prevot, A. S. H., Dommen, J., Duplissy, J., Metzger, A., Baltensperger, U., and Jimenez, J. L.: O/C and OM/OC ratios of primary, secondary, and ambient organic aerosols with highresolution time-of-flight aerosol mass spectrometry, *Environmental Science & Technology*, 42, 4478–4485, doi:10.1021/es703009q, 2008.
- Chhabra, P. S., Ng, N. L., Canagaratna, M. R., Corrigan, A. L., Russell, L. M., Worsnop, D. R., Flagan, R. C., and Seinfeld, J. H.: Elemental composition and oxidation of chamber organic aerosol, Atmospheric Chemistry and Physics, 11, 8827–8845, doi:10.5194/acp-11-8827-2011, 2011.
- Dillner, A. M. and Takahama, S.: Predicting ambient aerosol thermal-optical reflectance (TOR) measurements from infrared spectra: organic carbon, Atmospheric Measurement Techniques, 8, 1097–1109, doi:10.5194/amt-8-1097-2015, 2015a.

Dillner, A. M. and Takahama, S.: Predicting ambient aerosol thermal-optical reflectance measurements from infrared spectra: elemental carbon, *Atmospheric Measurement Techniques*, 8, 4013–4023, doi:10.5194/amt-8-4013-2015, 2015b.

- Haaland, D. M. and Thomas, E. V.: Partial least-squares methods for spectral analyses. 1. Relation to other quantitative calibration methods and the extraction of qualitative information, *Analytical Chemistry*, 60, 1193–1202, doi:10.1021/ac00162a020, 1988.
- Kvalheim, O. M., Arneberg, R., Bleie, O., Rajalahti, T., Smilde, A. K., and Westerhuis, J. A.: Variable importance in latent variable regression models, *Journal of Chemometrics*, 28, 615– 622, doi:10.1002/cem.2626, 2014.
- Reggente, M., Dillner, A. M., and Takahama, S.: Predicting ambient aerosol thermal-optical reflectance (TOR) measurements from infrared spectra: extending the predictions to different years and different sites, *Atmospheric Measurement Techniques*, 9, 441–454, doi:10.5194/ amt-9-441-2016, 2016.
- Ruggeri, G., Bernhard, F. A., Henderson, B. H., and Takahama, S.: Model-measurement comparison of functional group abundance in α-pinene and 1,3,5-trimethylbenzene secondary organic aerosol formation, Atmospheric Chemistry and Physics Discussions, 2016, 1–34, doi: 10.5194/acp-2016-46, 2016.
- Ruthenburg, T. C., Perlin, P. C., Liu, V., McDade, C. E., and Dillner, A. M.: Determination of organic matter and organic matter to organic carbon ratios by infrared spectroscopy with application to selected sites in the IMPROVE network, *Atmospheric Environment*, 86, 47–57, doi:10.1016/j.atmosenv.2013.12.034, 2014.
- Takahama, S. and Dillner, A. M.: Model selection for partial least squares calibration and implications for analysis of atmospheric organic aerosol samples with mid-infrared spectroscopy, *Journal of Chemometrics*, 29, 659–668, doi:10.1002/cem.2761, 2015.

---

## Author Comment (AC2) · 14 Jun 2016

**Response to Reviewer #2**

In this well-structured and very detailed paper, four sparse algorithms (SPLSa, SPLSb, EN and EN-PLS) were considered and evaluated for selecting relevant mid-infrared absorption bands in the calibration model building process. Using FTIR spectra obtained in transmission mode, two types of sparse calibration models were constructed for predicting/interpreting: (1) abundances of four organic functional groups (alcohol COH, carboxylic COH, alkane CH and carbonyl CO) and (2) TOR OC and EC concentrations in ambient aerosol samples. The paper has also presented a thorough analysis of the constructed models. I find the paper suitable for publication in AMT and only have very few minor comments[.]

> We thank the reviewer for the positive comments.

Page 3, line 75 onwards: Samples were taken from seven sites in the IMPROVE network. Did the investigators set out any criteria for selecting these sites for the study? It might be a good idea to provide a brief description of the sites to gauge their representativeness of the network. Throughout the paper, nothing was mentioned about these sites except that they were comprised of rural and urban sites (from Figures 3 and 4).

> These sites comprise the entire IMPROVE 2011 spectra set available (Ruthenburg et al., 2014), and we have developed and extensively evaluated a set of base case models for prediction of FG and TOR OC and EC Ruthenburg et al. (2014); Takahama and Dillner (2015); Dillner and Takahama (2015a,b). The number of monitoring sites for which FT-IR is available has since expanded (e.g., Reggente et al., 2016), but for this work we have selected the original, well-studied set of samples to specifically investigate the impact of sparse algorithms on each type of calibration model.

> We have added to the Introduction section (1):

> "These past studies evaluate various performance metrics achieved by statistical calibration models using the full set of wavenumbers, and we evaluate the effect of variable selection on model performance and interpretation."

> The chemical composition as parameterized by OM/OC has been included in the Methods section (2.1.1):

> "The OM/OC ratio estimated in ambient samples span a range of 1.46 and 2.01 between the 10th and 90th percentiles, with a median ratio of 1.69 (Ruthenburg et al., 2014)."

> We are currently extending our application of sparse algorithms for further understanding $PM_{2.5}$ in a wide range of environments. With regards to representativeness, we hope to address this topic in future studies.

Page 4, line 100: The 250 laboratory standards used were mixtures of seven compound types. What are these compounds? If this has been described elsewhere, citing the relevant literature would suffice.

> The compounds are: 1-docosanol, D-glucose, fructose, levoglucosan, malonic acid, adipic acid, suberic acid, arachidyl dodecanoate, 12-tricosanone. We apologize for the error but

there are nine compounds instead of seven, and has been corrected. These compounds have been documented by Ruthenburg et al. (2014) and the citation has been inserted in to the Methods section (2.1.1).

Page 40, Figure 3 legend: What do the investigators mean by anomalous clusters? PM: $PM_{10}$ or $PM_{2.5}$?

We thank the reviewer for pointing out these omissions. In the caption of Figure 3, we have added this statement: ""Anomalous" samples are those identified by Ruthenburg et al. (2014) (38 samples or 5% of the total set) that share similar spectral profiles and large disagreement with TOR in estimated OC. The cause for the disagreement is at present time unknown."

We have noted that these are $PM_{2.5}$ samples in the Introduction section (1): "We revisit calibration models for four FGs developed using laboratory standards (Ruthenburg et al., 2014; Takahama and Dillner, 2015), and TOR OC and EC calibration models developed with ambient $PM_{2.5}$ samples collected in 2011 at seven sites within the Interagency Monitoring of PROtected Visual Environment (IMPROVE; Malm et al., 1994; Hand et al., 2012) monitoring network (Dillner and Takahama, 2015a,b)."

and in Methods section (2.1.1): "For this work, we use 794 pairs of ambient $PM_{2.5}$ samples collected in the IMPROVE monitoring network[...]"

Grammatical/typographical corrections: Page 38, Figure 1 caption: Should this be Appendix C instead of Section C? Please go over the manuscript again and proofread it.

We thank the reviewer for catching this and other typographical errors. This is Appendix C instead of Section C and the correction has been made. Additionally, minor errors found upon final proofreading have been corrected and are highlighted in the manuscript accompanying this response.

**References**

Dillner, A. M. and Takahama, S.: Predicting ambient aerosol thermal-optical reflectance (TOR) measurements from infrared spectra: organic carbon, *Atmospheric Measurement Techniques*, 8, 1097–1109, doi:10.5194/amt-8-1097-2015, 2015a.

Dillner, A. M. and Takahama, S.: Predicting ambient aerosol thermal-optical reflectance measurements from infrared spectra: elemental carbon, *Atmospheric Measurement Techniques*, 8, 4013–4023, doi:10.5194/amt-8-4013-2015, 2015b.

Hand, J. L., Schichtel, B. A., Pitchford, M., Malm, W. C., and Frank, N. H.: Seasonal composition of remote and urban fine particulate matter in the United States, *Journal of Geophysical Research: Atmospheres*, 117, doi:10.1029/2011JD017122, 2012.

Malm, W. C., Sisler, J. F., Huffman, D., Eldred, R. A., and Cahill, T. A.: Spatial and seasonal trends in particle concentration and optical extinction in the United States, *Journal of Geophysical Research: Atmospheres*, 99, 1347–1370, doi:10.1029/93JD02916, 1994.

Reggente, M., Dillner, A. M., and Takahama, S.: Predicting ambient aerosol thermal-optical reflectance (TOR) measurements from infrared spectra: extending the predictions to different years and different sites, *Atmospheric Measurement Techniques*, 9, 441–454, doi:10.5194/amt-9-441-2016, 2016.

Ruthenburg, T. C., Perlin, P. C., Liu, V., McDade, C. E., and Dillner, A. M.: Determination of organic matter and organic matter to organic carbon ratios by infrared spectroscopy with application to selected sites in the IMPROVE network, *Atmospheric Environment*, 86, 47–57, doi:10.1016/j.atmosenv.2013.12.034, 2014.

Takahama, S. and Dillner, A. M.: Model selection for partial least squares calibration and implications for analysis of atmospheric organic aerosol samples with mid-infrared spectroscopy, *Journal of Chemometrics*, 29, 659–668, doi:10.1002/cem.2761, 2015.

---

## Author Comment (AC3) · 14 Jun 2016

[revised manuscript text omitted]

**2.1 Experimental methods and spectra processing**

**2.1.1 Laboratory and ambient samples**

For this work, we use 794 pairs of ambient $PM_{2.5}$ samples collected in the IMPROVE monitoring network, 250 laboratory standards, and 54 blank samples used previously by Ruthenburg et al. (2014), Dillner and Takahama (2015a, b), and Takahama and Dillner (2015) for building FG and TOR OC and EC calibration models with canonical PLS regression. A pair of ambient samples consists of particles collected on 25 mm quartz fiber filters and 25 mm PTFE filters. The quartz filters are analyzed by Total Optical Reflectance (TOR) IMROVE_A protocol for OC and EC mass (Chow et al., 2007), and the PTFE filters are used for acquisition of infrared spectra, among other properties. Monthly median values of OC loadings in blank samples are subtracted from ambient TOR OC loadings to account for the gas phase adsorption artifact by quartz fiber filters (Dillner and Takahama, 2015a). The OM/OC ratio estimated in ambient samples span a range of 1.46 and 2.01 between the 10th and 90th percentiles, with a median ratio of 1.69 (Ruthenburg et al., 2014). 
[revised manuscript text omitted]

are constructed identically to the most accurate class of full wavenumber models described previously (Ruthenburg et al., 2014; Takahama and Dillner, 2015; Dillner and Takahama, 2015a, b). For FG calibration, 158 laboratory standards are used for the calibration set while 80 similar laboratory samples and all 794 ambient samples are reserved for the test set (Takahama and Dillner, 2015). No blank samples are included in the FG calibration, though samples with particular FG concentrations

5 of zero (e.g., compounds that only contain other FGs than those for which the calibration model is being built) are included (Ruthenburg et al., 2014). For TOR OC and EC calibration, the 794 ambient samples are arranged in order of TOR reference concentration and every third sample is selected for the test set and the remaining two-thirds of samples used for the calibration set (Dillner and Takahama, 2015a, b). Similarly, one-third of blank samples are reserved for the test set while the remaining two-thirds are included in the calibration set. While the division between sets is arbitrary in the TOR OC case, for TOR EC

10 the blanks are first arranged according to their predicted concentrations using a calibration model developed without blank samples (Dillner and Takahama, 2015b) and every third selected for the test set as for ambient samples. Dillner and Takahama (2015b) additionally divided the EC samples into high and low concentration samples to build a hybrid (piecewise) calibration model for improving predictions for the latter group of samples. For the purpose of this work, we only consider a single calibration model that spans the entire range of concentrations for each algorithm and spectra preparation. Full wavenumber

15 calibration models for TOR OC and EC developed using this protocol have demonstrated ability to capture overall variations in concentrations which span a wide range of PM composition and environmental conditions (Dillner and Takahama, 2015a, b).

[revised manuscript text omitted]
 (e.g., stratified by environmental conditions or chemical composition). This work provides a demonstration of how molecular structure can be associated with other quantifiable metrics of complex PM to which spectral features from FT-IR can be correlated.

**Appendix A: Notation**

Tables A1 and A2 summarize notation used for matrices and vectors with their corresponding dimensions. Matrices in written

10    in uppercase italic bold and vectors in lowercase italic bold. Vectors are column vectors by convention; row vectors are written as transposed vectors.

**Table A1.** Dimensions and indexing variables.

| Scalar variable | Description | Dummy index |
|:---:|:---|:---:|
| $N$ | number of samples | $i$ |
| $M$ | number of independent variables (wavenumbers) | $j$ |
| $K$ | number of latent variables used | $h, k$ |

**Table A2.** Arrays and dimensions.

| Array variable | Vector/scalar notation | Description |
|:---:|:---:|:---|
| $\boldsymbol{X}$ | $[\boldsymbol{x}_i^T]$ | matrix of spectra ($N \times M$) |
| $\boldsymbol{Y}$ | $[\boldsymbol{y}_k]$ | matrix of dependent variables ($N \times 1$) |
| $\boldsymbol{B}$ | $[\boldsymbol{b}_k]$ | matrix of PLS coefficients ($M \times 1$) |
| $\boldsymbol{T}$ | $[\boldsymbol{t}_h]$ | matrix of X scores ($N \times K$) |
| $\boldsymbol{P}$ | $[\boldsymbol{p}_h]$ | matrix of X loadings ($M \times K$) |
| $\boldsymbol{E}_X$ | $[\boldsymbol{e}_{x,i}^T]$ | matrix of X residuals ($N \times M$) |
| $\boldsymbol{Q}$ | $[\boldsymbol{q}_h]$ | matrix of Y loadings ($1 \times K$) |
| $\boldsymbol{E}_Y$ | $[\boldsymbol{e}_{y,i}^T]$ | matrix of Y residuals ($N \times 1$) |
| $\boldsymbol{R}$ | $[\boldsymbol{r}_h]$ | matrix of X direction vectors ($M \times K$) |
| $\boldsymbol{W}$ | $[\boldsymbol{w}_h]$ | matrix of X weights ($M \times K$) |

**Appendix B: Model specification**

The derivation, properties, and implementation of sparse methods used in this manuscript are described in detail by their respective authors: SPLSa (Lê Cao et al., 2008), SPLSb (Chun and Keles, 2010), EN (Zou and Hastie, 2005; Friedman et al., 2010), and EN-PLS (Fu et al., 2011). In this section, we briefly summarize the methods using consistent notation such that 1) their problem statements can be compared through their objective functions and constraints (formulated as penalties), and 2) how sparsity is controlled by their respective tuning parameters is apparent. An overview of methods and the parameters over which models are explored is provided in Table B1. For PLS-methods, the more general case for multivariate $\boldsymbol{Y}$ is introduced, and specific simplifications for univariate $\boldsymbol{y}$ is described where notable. For solving the PLS problem, we use the NIPALS algorithm in each case as the weight vectors derived from this algorithm can be used for calculating VIP scores.

**B1 Partial Least Squares (PLS)**

A search for LVs can be framed as an optimization problem to maximize covariance between response and explanatory variables under a set of transformations (Burnham et al., 1996; Chun and Keles, 2010; Lee et al., 2011; Filzmoser et al., 2012; Liu, 2014). Writing the matrix product of the spectra and response variables as $\boldsymbol{Z} = \boldsymbol{X}^T \boldsymbol{Y}$, the transformations are introduced through the weight vector $\boldsymbol{w}$ for each $k$th LV:

$$
\begin{aligned}
\operatorname*{arg\,max}_{\boldsymbol{w}_k} \quad & \boldsymbol{w}_k^T \boldsymbol{Z}_k \boldsymbol{Z}_k^T \boldsymbol{w}_k \\
\text{s.t.} \quad & \|\boldsymbol{w}_k\|^2 = 1
\end{aligned}
\tag{B1}
$$

with a constraint to ensure that the weight vectors are normalized. In the non-linear iterative partial least squares (NI-PALS) algorithm (Wold, 1966; Martens, 1991), the weight vectors are calculated from the deflated (residual) matrix $\boldsymbol{Z}_k = \boldsymbol{X}_k^T \boldsymbol{Y}_k$ obtained from the $k$th iteration (Burnham et al., 1996; Chun and Keles, 2010; Lee et al., 2011) in which $\boldsymbol{X}_k = (\boldsymbol{I}_N - \boldsymbol{T}_{k-1} \boldsymbol{T}_{k-1}^+) \boldsymbol{X}_{k-1}$ and $\boldsymbol{Y}_k = (\boldsymbol{I}_N - \boldsymbol{T}_{k-1} \boldsymbol{T}_{k-1}^+) \boldsymbol{Y}_{k-1}$. $\boldsymbol{I}_N$ is the identity matrix of dimension $N \times N$, $\boldsymbol{T}_{k-1} = [\boldsymbol{X}_1 \boldsymbol{w}_1, \boldsymbol{X}_2 \boldsymbol{w}_2, \ldots, \boldsymbol{X}_{k-1} \boldsymbol{w}_{k-1}]$ and $\boldsymbol{T}^+$ is the Moore-Penrose inverse of $\boldsymbol{T}$. $\boldsymbol{T}_0 \equiv \boldsymbol{0}_{N \times K}$ such that $\boldsymbol{X}_0 = \boldsymbol{X}$ and $\boldsymbol{Y}_0 = \boldsymbol{Y}$ (Burnham et al., 1996; ter Braak and de Jong, 1998; Lee et al., 2011). The weight vectors correspond to eigenvectors of $\boldsymbol{Z}_k \boldsymbol{Z}_k^T$ (Höskuldsson, 1988; Rosipal and Krämer, 2006).

$\boldsymbol{w}$ and $\boldsymbol{r}$ introduced as column elements of $\boldsymbol{W}$ and $\boldsymbol{R}$, respectively, in Section 2.2 are related in concept and often referred to as loading weights, loadings, weights, and direction vectors interchangeably (e.g., Haaland and Thomas, 1988; Kvalheim and Karstang, 1989; Mevik and Wehrens, 2007; Lê Cao et al., 2008; Chun and Keles, 2010; Lee et al., 2011; Filzmoser et al., 2012). In this manuscript, we adopt the convention of referring to $\boldsymbol{w}$ as (loading) weights and $\boldsymbol{r}$ as direction vectors, respectively. Using the definition of deflated matrices, we can also write the relationship between the loading weights and direction vectors as $\boldsymbol{X}_k \boldsymbol{w}_k = \boldsymbol{X} \boldsymbol{r}_k = \boldsymbol{t}_k$, and $\boldsymbol{r}_k = (\boldsymbol{I}_M - \boldsymbol{R} \boldsymbol{P}^T) \boldsymbol{w}_k$ where $\boldsymbol{I}_M$ is the identity matrix of dimensions $M \times M$ (ter Braak and de Jong, 1998). The reader will note that $\boldsymbol{Z}$ is proportional to the cross-covariance matrix between $\boldsymbol{X}$ and $\boldsymbol{Y}$, and the objective function (Equation B1) is in fact proportional to the inner product of the cross covariances between $\boldsymbol{Y}_k$ and the transformed variables $\boldsymbol{t}_k$ for each LV $k$.

To solve for the underlying weights and direction vectors which satisfy these equations, we use the NIPALS algorithm implemented in the `pls` library (Mevik and Wehrens, 2007) for the R programming language (R Core Team, 2014) in this work. Candidate models are generated by varying $K = \{1, 2, \ldots, 120\}$, from which one is selected by penalizing model variance over an ensemble of scaling factors and combining them through consensus scoring (Takahama and Dillner, 2015).

**5    B2    Elastic Net (EN) regularization**

EN regularization is not a variant of PLS but solves for regression coefficients in Equation 1 without using LVs. The objective function to be minimized is similar to the residual sum-of-squares (RSS) used in ordinary least squares regression, but with additional constraints imposed on the regression vector (Zou and Hastie, 2005):

$$\underset{\boldsymbol{b}}{\arg\min} \|\boldsymbol{y} - \boldsymbol{X}\boldsymbol{b}\|_2^2 + \lambda_1 \|\boldsymbol{b}\|_1 + \lambda_2 \|\boldsymbol{
[revised manuscript text omitted]

[Figure]

**Figure 4.** Comparison statistics of full and reduced wavenumber solutions show sensitivity of model predictions to sparsity. OC and EC correspond to predictions from direct calibration to TOR measurements rather than summing FGs.

[Figure]

**Figure 5.** Predictions vs. reference for ambient samples (direct calibration).

[Figure]

**Figure 6.** VIP for solutions using raw spectra. Dark gray lines in the "full" solution panels correspond to the first loading weights, and the vertical bars in every panel extend from zero to VIP scores. Red points accompanying vertical bars indicate wavenumbers for which regression coefficients are positive and blue points indiciate wavenumbers for which coefficients are negative. Regions up to VIP scores of 0.5 are shaded to indicate VIP scores not considered for our interpretation (Section 3.3).

[Figure]

**Figure 7.** VIP for solutions using baseline corrected spectra. Lines and colors are as indicated for Figure 6.

[Figure]

**Figure 8.** Example group of baseline corrected spectra shown with VIP scores above 0.5 overlayed. The VIP is derived from all calibration spectra consisting of laboratory standards (same as those shown in Figure 7). The column of panels on right show sensitivity of predictions for each FG for this subset of samples.